# Genome-wide functional analysis of phosphatases in the pathogenic fungus *Cryptococcus neoformans*

Jae-Hyung Jin [1,7], Kyung-Tae Lee [1,7], Joohyeon Hong[1], Dongpil Lee[1], Eun-Ha Jang[1], Jin-Young Kim [1], Yeonseon Lee[1], Seung-Heon Lee [1], Yee-Seul So[1], Kwang-Woo Jung [1,6], Dong-Gi Lee [1], Eunji Jeong[1], Minjae Lee [1], Yu-Byeong Jang [1], Yeseul Choi [1], Myung Ha Lee [1], Ji-Seok Kim [1], Seong-Ryong Yu [1], Jin-Tae Choi [1], Jae-Won La[1], Haneul Choi[1], Sun-Woo Kim[1], Kyung Jin Seo[1], Yelin Lee [1], Eun Jung Thak[2], Jaeyoung Choi [3], Anna F. Averette[4], Yong-Hwan Lee [5], Joseph Heitman[4], Hyun Ah Kang[2], Eunji Cheong [1] & Yong-Sun Bahn [1✉]

Phosphatases, together with kinases and transcription factors, are key components in cellular signalling networks. Here, we present a systematic functional analysis of the phosphatases in *Cryptococcus neoformans*, a fungal pathogen that causes life-threatening fungal meningoencephalitis. We analyse 230 signature-tagged mutant strains for 114 putative phosphatases under 30 distinct in vitro growth conditions, revealing at least one function for 60 of these proteins. Large-scale virulence and infectivity assays using insect and mouse models indicate roles in pathogenicity for 31 phosphatases involved in various processes such as thermotolerance, melanin and capsule production, stress responses, *O*-mannosylation, or retromer function. Notably, phosphatases Xpp1, Ssu72, Siw14, and Sit4 promote blood-brain barrier adhesion and crossing by *C. neoformans*. Together with our previous systematic studies of transcription factors and kinases, our results provide comprehensive insight into the patho-biological signalling circuitry of *C. neoformans*.

[1] Department of Biotechnology, College of Life Science and Biotechnology, Yonsei University, Seoul 03722, Korea. [2] Department of Life Science, College of Natural Science, Chung-Ang University, Seoul 06974, Korea. [3] Smart Farm Research Center, Korea Institute of Science and Technology, Gangneung 25451, Korea. [4] Departments of Molecular Genetics and Microbiology, Medicine, and Pharmacology and Cancer Biology, Duke University Medical Center, Durham, NC 27710, USA. [5] Department of Agricultural Biotechnology, Seoul National University, Seoul 08826, Korea. [6] Present address: Radiation Research Division, Korea Atomic Energy Research Institute, Jeongeup 56212, Korea. [7] These authors contributed equally: Jae-Hyung Jin, Kyung-Tae Lee.
✉email: ysbahn@yonsei.ac.kr

A ll living organisms employ complex signalling pathways for dynamic response to changes in their environment to survive and proliferate. When an environmental cue occurs, an organism receives the signal through a sensor (i.e., receptor protein), which subsequently activates downstream effectors to counteract the incoming stress. Upon resolution of or adaptation to the cue, the signalling pathway is generally desensitized in a timely manner in order to be reactivated later. This process of desensitization is governed by phosphorylation and dephosphorylation, events modulated by kinases and phosphatases, respectively, which constitute key post-translational modifications that switch signalling components on or off. Thus, orchestrated regulation of kinases and phosphatases in signalling pathways are critical to maintaining cellular homoeostasis[1].

Fungal pathogens also utilize these complex signalling pathways to respond and adapt to dramatic environmental changes occurring during infection, colonization, proliferation, and dissemination to various tissues within a host. Although several key signalling pathways governing fungal pathogenicity have been elucidated in past decades[2], recent large-scale functional analyses of fungal kinases and transcription factors (TFs) provide even more comprehensive insights into how fungal signalling pathways modulate infection and virulence. Indeed, 178 TFs and 183 kinases were identified in *Cryptococcus neoformans*, which causes life-threatening meningoencephalitis mainly in immunocompromised patients and is responsible for more than 180,000 deaths annually[3]; of the 155 and 129, respectively, functionally characterized in vitro and in vivo[4,5], 45 TFs and 63 kinases are involved in its pathogenicity. Nevertheless, the regulation of these signalling components and their coordination in pathogenicity remain elusive.

To better understand TF and kinase networks, co-analysis of the phosphatase networks as signalling counterparts is essential. Historically, phosphatases have received less attention than kinases, probably because kinases, which possess high substrate specificity, are generally considered to be better drug targets than substrate-promiscuous phosphatases[6]. However, recent systematic functional phosphatome data in several fungal pathogens indicate that phosphatases play critical roles in maintaining cellular homoeostasis by controlling growth and cell cycle, differentiation, stress response, and metabolism. In ascomycete fungi, genome-wide analyses of phosphatases have identified 32 protein phosphatases and functionally characterized 24 of them in *Aspergillus fumigatus*, the majority of which contribute to stress response, iron assimilation, and toxin production and resistance[7]. Furthermore, in the wheat scab fungus *Fusarium graminearum*, Yun et al. identified 82 phosphatase genes, disrupted 71 of them, and functionally analysed these mutants across 15 phenotypic traits (e.g., growth, nutrient response, and virulence), finding that 25 of these phosphatases are involved in virulence of the phytopathogenic fungus[8]. Among phosphatases, calcineurin is a well-established drug target and a globally conserved fungal virulence factor[9].

The goal of this study was to systematically analyse the functions of the *C. neoformans* phosphatase network and interplay with its kinase and TF networks to better understand its pathobiological signalling. To this end, we constructed a high-quality library of 219 signature-tagged gene-deletion mutant strains representing 109 phosphatases out of 139 putative phosphatases identified in *C. neoformans*, in addition to 11 signature-tagged mutants representing six phosphatases that we previously constructed[4,5,10,11]. Using a total of 230 signature-tagged mutants representing 114 phosphatases, we analysed their phenotypic traits under 30 distinct in vitro conditions and performed a large-scale virulence assay using two model host systems (insect and murine). This entire phosphatase phenome data set is freely available to the public through the *C. neoformans* Phosphatase Phenome Database (http://phosphatase.cryptococcus.org).

## Results

**Identification of phosphatases in *C. neoformans*.** To select putative phosphatase genes, we surveyed a curated annotation of phosphatases in the FungiDB *C. neoformans* (H99 strain) genome database (http://fungidb.org/fungidb) and validated the presence of phosphatase-related domains using protein sequence analysis and classification (Interpro). Through these analyses, we retrieved a total of 139 putative phosphatase genes in *C. neoformans* (Fig. 1a, b and Supplementary Data 1). We made three notable findings. First, phosphatases are generally less evolutionarily conserved than kinases, but more conserved than TFs (Supplementary Fig. 1 and Supplementary Data 2). Second, despite a paucity of tyrosine kinases [only three tyrosine kinase-like (TKL) proteins], *C. neoformans* contains 21 protein tyrosine phosphatases (PTPs). Third, compared to other non-pathogenic and pathogenic yeasts, *C. neoformans* contains a similar number of putative phosphatases (Fig. 1c and Supplementary Data 3).

**Construction of the *C. neoformans* phosphatase mutant library.** To obtain comprehensive insights into the *C. neoformans* phosphatome networks and their biological functions, we constructed gene-deletion mutants for each phosphatase gene and analysed their in vitro and in vivo phenotypic traits. Among the 139 putative phosphatase genes, 15 have been functionally characterized previously by gene-deletion studies (*PTP1*[10], *PTP2*[10], *YSA1*[12], *CNA1*[13], *CAC1*[11], *TPS2*[14], *CCR4*[15], *HAD1*[16], *EPP1*[17], *XPP1*[17], *APH1*[18], *ASP1*[19], *ISC1*[20], *PPG1*[21], and *PPH3*[21]). In addition, we previously deleted two phosphatase genes with kinase domains (*OXK1* and *FBP26*) for construction of the kinase mutant library[5] and one phosphatase gene with a DNA-binding domain (*APN2*) for the TF mutant library[4]. Besides the 11 signature-tagged mutant strains (two for each *PTP1*, *PTP2*, *OXK1*, *FBP26*, and *APN2* and one for *CAC1*) that we previously constructed, we performed a large-scale homologous recombination-based gene deletion using nourseothricin-resistance markers containing a series of unique oligonucleotide signature tags that we previously employed for construction of the TF and kinase mutant libraries[4,5] (Supplementary Data 4). To obtain a high-quality phosphatase mutant library, we constructed more than two independent mutants for each gene and verified their genotypes by diagnostic PCR and Southern blot analysis. As a result, here, we report 219 new mutant strains representing 109 phosphatases and the analysis of a total of 230 mutant strains representing 114 phosphatases (Supplementary Data 5). Disruption strategies, primer sequences, Southern blot results, and mutant phenome data are available in the *Cryptococcus neoformans* Phosphatase Phenome Database (http://phosphatase.cryptococcus.org) we constructed for this study. For the remaining 25 phosphatase genes, we failed to obtain any viable transformants or obtained only potential aneuploid mutants possessing both wild-type and mutant alleles after repeated attempts, suggesting these genes may be essential (Supplementary Data 6). We also developed the *Cryptococcus neoformans* Phenome Gateway Database (http://www.cryptococcus.org/), in which every TFs, kinases, and phosphatases studied[4,5] are linked to the most widely used fungal genome database "FungiDB" (https://fungidb.org/fungidb/) to maximize connectivity between research data.

**Phenotypic and in vivo expression profiling of phosphatome.** To elucidate the functions of *C. neoformans* phosphatases deleted in this and previous studies, we examined phenotypic traits under

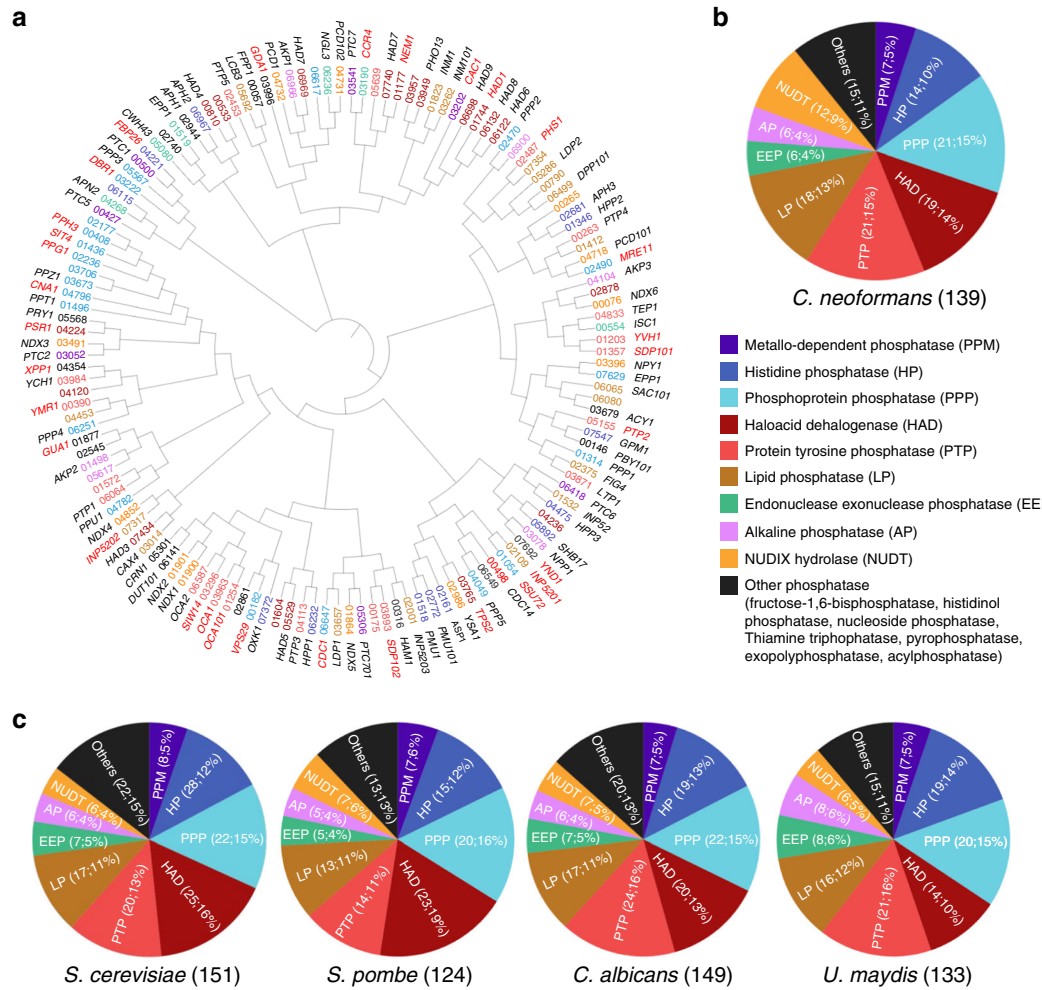

**Fig. 1 Classification of phosphatases in *C. neoformans* and other fungal species. a** Phylogenetic analysis of phosphatases utilized a protein sequence-based alignment in ClustalX2 (Science Foundation Ireland, Dublin, Ireland). These alignment data were used to illustrate the phylogenetic tree through a web-based drawing application (Interactive Tree Of Life, https://itol.embl.de). Red letters represent pathogenicity-related phosphatases in *C. neoformans* identified from 114 phosphatase gene-deletion mutants that we constructed. **b** Pie chart indicating the classification and distribution of 139 phosphatases based on phosphatase-related domain, as analysed by using the InterPro Database. **c** Pie chart indicating the distribution of putative phosphatase genes in the fungal species *S. cerevisiae*, *S. pombe*, *C. albicans*, and *U. maydis* surveyed and classified using the same strategies as for *C. neoformans*.

30 distinct in vitro conditions: growth at different temperatures (25, 30, 37, and 39 °C), mating efficiency, virulence factor production (capsule, melanin, and urease), stress responses (osmotic/cation salt, oxidative, genotoxic, ER, cell membrane/wall, and heavy metal stresses), and antifungal drug susceptibility (Supplementary Data 7). The whole phenome data set of the phosphatase mutant collection was qualitatively illustrated with a colour scale (Fig. 2 and Supplementary Data 7). This systematic phenotypic analysis revealed that ~53% of the phosphatase mutants (60/114) showed at least one discernible phenotype (Fig. 2); 72% of these (43/60) have not been previously functionally analysed. When we compared our phosphatase phenome data with their corresponding BLAST matrix data (Supplementary Fig. 2 and Supplementary Data 2), we found that the putative essential phosphatases that we could not disrupt and the phosphatases showing multiple phenotypic traits were generally more evolutionarily conserved. Phenotypic clustering of phosphatase mutants revealed groups of phosphatases that could be directly or indirectly correlated with regards to cellular function (Fig. 2).

In addition, we monitored how each of the 139 phosphatase genes were transcriptionally regulated during murine infection by assessing expression levels of each phosphatase gene in recovered tissues (lungs, brain, spleen, and kidneys) after 3, 7, 14, and 21 days post-infection (dpi) with strain H99S using the nCounter gene expression profile (NanoString, Seattle, WA, USA) with 139 novel phosphatase probes. The in vivo expression levels of each phosphatase were normalized to average in vivo expression levels of eight housekeeping genes (mitochondrial protein, CNAG_00279; microtubule binding protein, CNAG_00816; aldose reductase, CNAG_02722; cofilin, CNAG_02991; actin, CNAG_00483; tubulin β chain, CNAG_01840; tubulin α-1A chain, CNAG_03787; histone H3, CNAG_04828) and compared to those under basal growth conditions [yeast extract-peptone-dextrose (YPD) medium at 30 °C] (Fig. 2, Supplementary Fig. 3, and Data 8). Notably, we detected increased in vivo expression of numerous phosphatase genes in the brain, kidneys, and spleen during early infection (3–7 dpi; Fig. 2). Normally, cryptococcal CFUs are barely detected in the brain and other organs, except the lungs, during early infection stage in the intranasal inhalation infection model, partly due to the detection limit of the conventional fungal burden assay. However, we used the Nanostring nCounter platform, which can detect a single gene transcript without amplification[22,23]. Therefore, once a small number of *C. neoformans* cells are hematogenously disseminated

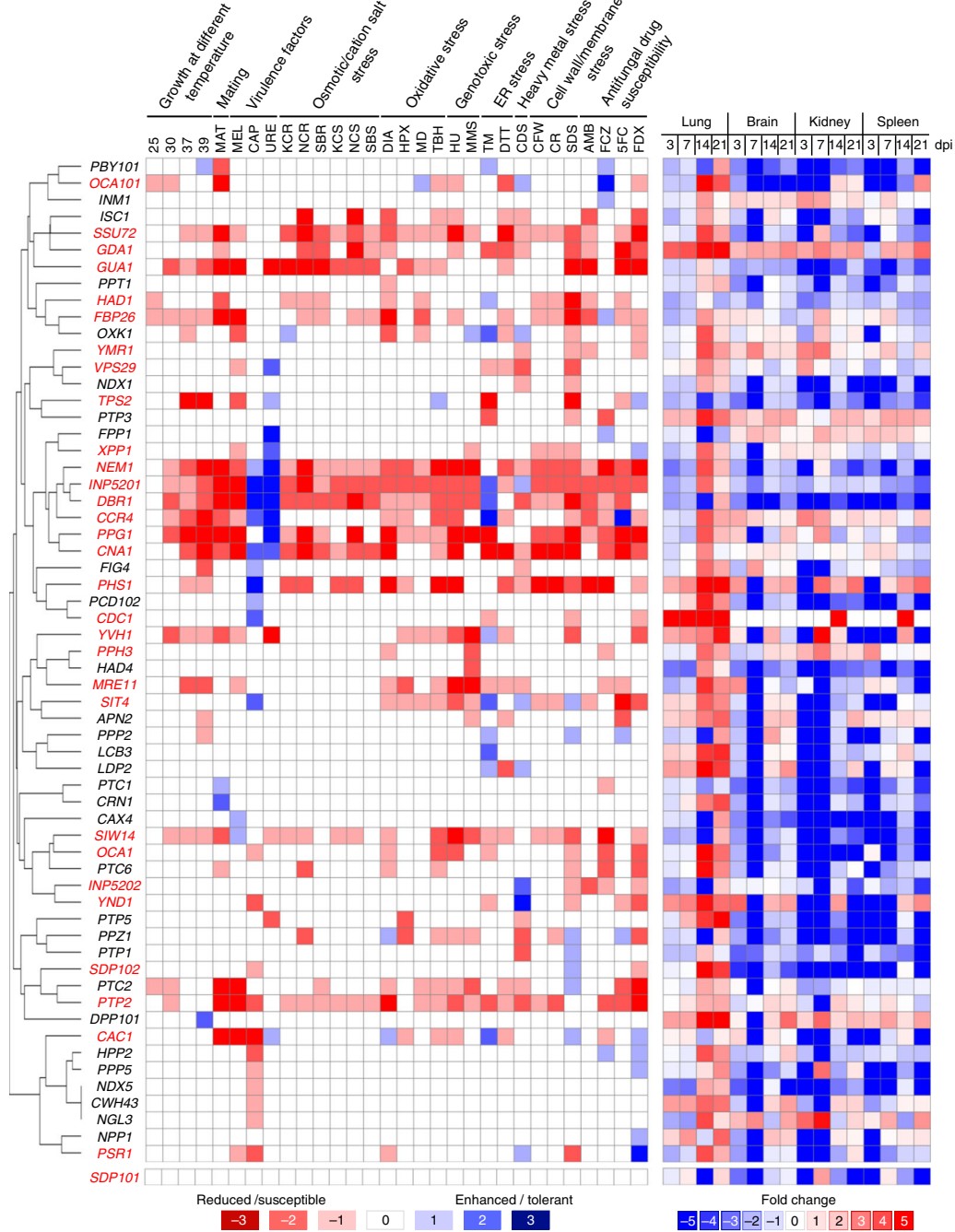

**Fig. 2 Phenotypic clustering and in vivo expression profiling of phosphatases in *C. neoformans*.** In vitro phenotypic traits were examined under 30 different growth conditions and scored on a 7-point scale (−3: strongly reduced/susceptible, −2: moderately reduced/susceptible, −1: weakly reduced/susceptible, 0: wild-type like, +1 weakly enhanced/tolerant, +2: moderately enhanced/tolerant, +3: strongly enhanced/tolerant). All phenotypic data are available in the *Cryptococcus neoformans* Phosphatase Phenome Database (http://phosphatase.cryptococcus.org). More than three biologically independent experiments were performed for each phenotypic trait. Hierarchical phenotypic clustering of 60 phosphatases showing at least one phenotypic trait was performed with one minus Pearson correlation in Morpheus (https://software.broadinstitute.org/morpheus). The right panel shows the corresponding in vivo gene expression profiles for each phosphatase gene determined by NanoString nCounter platform analysis during intranasal murine infection with *C. neoformans*. Red letters represent pathogenicity-related phosphatases. The *sdp101*Δ mutant, which did not show any in vitro phenotypes but exhibited reduced infectivity, was also included. *Abbreviations*: 25 25 °C, 30 30 °C, 37 37 °C, 39 39 °C, CAP capsule production, MEL melanin production, URE urease production, MAT mating, HPX hydrogen peroxide, TBH *tert*-butyl hydroperoxide, MD menadione, DIA diamide, MMS methyl methanesulphonate, HU hydroxyurea, 5FC 5-flucytosine, AMB amphotericin B, FCZ fluconazole, FDX fludioxonil, TM tunicamycin, DTT dithiothreitol, CDS cadmium sulfate, SDS sodium dodecyl sulfate, CR Congo red, CFW calcofluor white, KCR YPD + 1.5 M KCl, NCR YPD + 1.5 M NaCl, SBR YPD + 2 M sorbitol, KCS YP + 1 M KCl, NCS YP + 1 M NaCl, SBS YP + 2 M sorbitol.

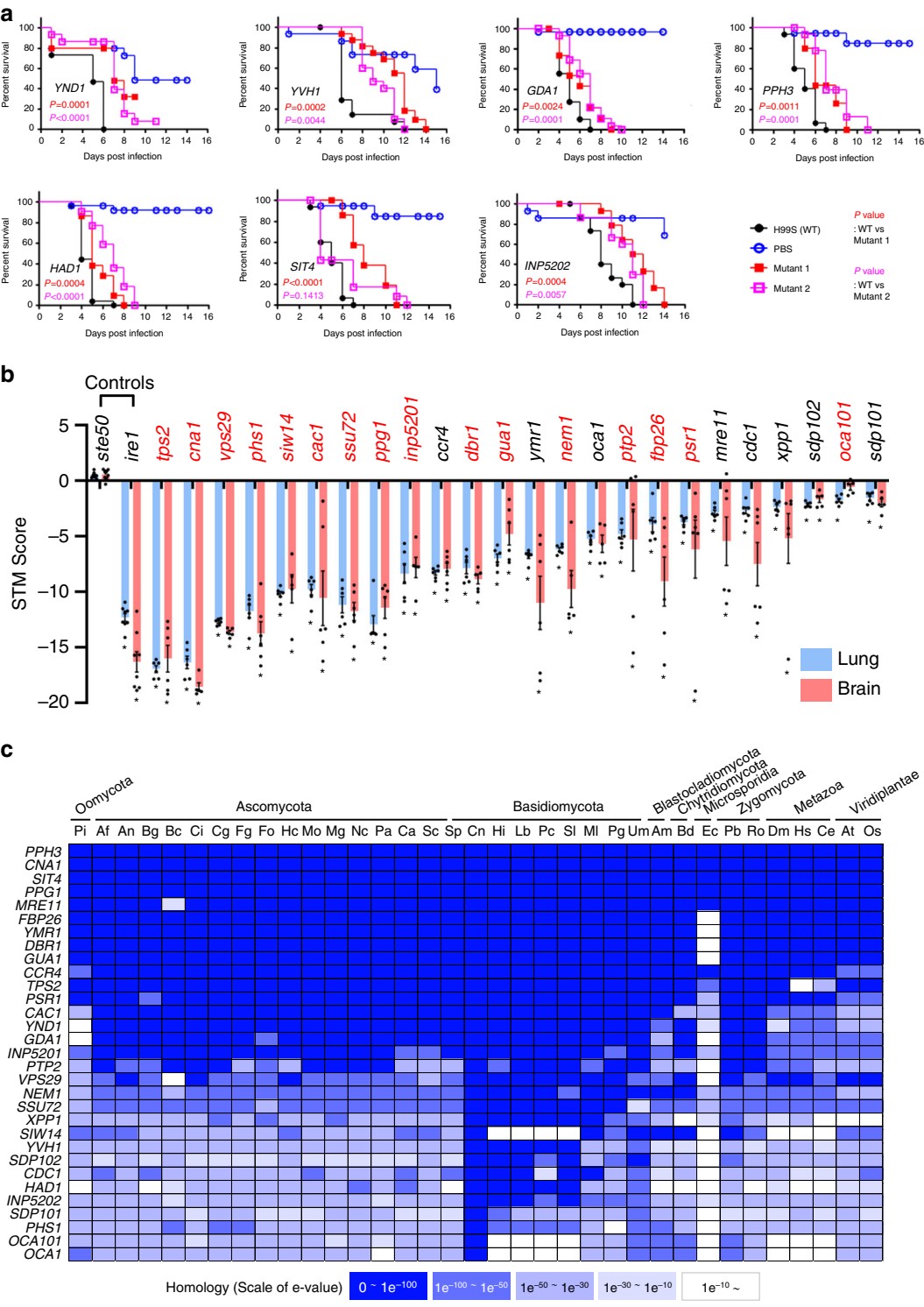

from the lungs to other organs during early infection, phosphatase transcripts can be detected 3–7 dpi.

**Phosphatases governing *C. neoformans* pathogenicity**. To identify phosphatases required for the pathogenicity of *C. neoformans*, we performed two large-scale infectivity and virulence assays: (1) a virulence assay using the insect larval model system *Galleria mellonella* and (2) signature-tagged mutagenesis (STM)-based lung and brain infectivity assays using a murine inhalation model. Both methods have been successfully used for identifying

virulence-related genes from a large-scale gene-deletion mutant set in previous studies[4,5,24].

Using the insect-killing virulence assay, we identified 23 phosphatase genes involved in virulence (Fig. 3a and Supplementary Fig. 4): *CNA1*, *TPS2*, *CAC1*, *PTP2*, *VPS29*, *PPH3*, *GDA1*, *YVH1*, *SSU72*, *PHS1*, *SIW14*, *DBR1*, *PSR1*, *YND1*, *INP5201*, *INP5202*, *HAD1*, *SIT4*, *PPG1*, *GUA1*, *NEM1*, *FBP26*, and *OCA101*. STM-based murine lung and brain infectivity revealed 24 infectivity-related phosphatase genes, *TPS2*, *CNA1*, *VPS29*, *PHS1*, *SIW14*, *CAC1*, *SSU72*, *PPG1*, *INP5201*, *CCR4*, *DBR1*, *GUA1*, *YMR1*, *NEM1*, *OCA1*, *PTP2*, *FBP26*, *PSR1*, *MRE11*, *CDC1*,

**Fig. 3 Pathogenicity-related phosphatases and their phenotypic traits in *C. neoformans*. a** Virulence-regulating phosphatases were identified by a *Galleria mellonella* insect killing assay ($n \geq 15$). *P* values were calculated using the log-rank (Mantel–Cox) test to measure statistical differences between the WT strain (H99S) and phosphatase mutants. **b** Infectivity-regulating phosphatases identified by the signature-tagged mutagenesis (STM)-based murine infectivity assay ($n = 6$, summary of Supplementary Fig. 5). STM scores were calculated by quantitative PCR. The *ste50Δ* and *ireΔ* mutants were used as virulent positive control and avirulent negative control strains, respectively. The statistically significant was calculated by one-way ANOVA analysis with Bonferroni's multiple comparison test. Data are presented as mean values ± standard error of mean (SEM). Red letters represent pathogenicity-related phosphatases identified by both murine and insect models. **c** BLAST matrix comparative search for pathogenicity-related phosphatases was performed using the Comparative Fungal Genomics Platform (http://cfgp.riceblast.snu.ac.kr). *Abbreviations*: Pi *Phytophthora infestans*, Af *Aspergillus fumigatus*, An *Aspergillus nidulans*, Bg *Blumeria graminis*, Bc *Botrytis cinerea*, Ci *Coccidioides immitis*, Cg *Colletotrichum graminicola*, Fg *Fusarium graminearum*, Fo *Fusarium oxysporum*, Hc *Histoplasma capsulatum*, Mo *Magnaporthe oryzae*, Mg *Mycosphaerella graminicola*, Nc *Neurospora crassa*, Pa *Podospora anserine*, Ca *Candida albicans*, Sc *Saccharomyces cerevisiae*, Sp *Schizosaccharomyces pombe*, Cn *Cryptococcus neoformans*, Hi *Heterobasidion irregular*, Lb *Laccaria bicolour*, Pc *Phanerochaete chrysosporium*, Sl *Serpula lacrymans*, Ml *Melampsora laricis-populina*, Pg *Puccinia graminis*, Um *Ustilago maydis*, Am *Allomyces macrogynus*, Bd *Batrachochytrium dendrobatidis*, Ec *Encephalitozoon cuniculi*, Pb *Phycomyces blakesleeanus*, Ro *Rhizopus oryzae*, Dm *Dorosophila melanogaster*, Hs *Homo sapiens*, Ce *Caenorhabditis elegans*, At *Arabidopsis thaliana*, Os *Oryza sativa*.

*XPP1*, *SDP102*, *SDP101*, and *OCA101* (Fig. 3b and Supplementary Fig. 5), 67% (16/24) of which overlapped with those identified by the insect-killing assay (*TPS2*, *CNA1*, *VPS29*, *PHS1*, *SIW14*, *CAC1*, *SSU72*, *PPG1*, *INP5201*, *DBR1*, *GUA1*, *NEM1*, *PTP2*, *FBP26*, *PSR1*, and *OCA101*; Fig. 3b, red). Among these, Cna1 (the catalytic subunit of calcineurin)[13], Cac1 (adenylyl cyclase)[11], Had1 (haloacid dehalogenase)[16], and Ptp2 (phosphotyrosine phosphatase)[10] have previously been reported as requisite for *C. neoformans* virulence, further corroborating our findings.

According to phenome data, 30 out of the 31 pathogenicity-related phosphatase mutants exhibited at least one phenotypic trait (Fig. 2 and Supplementary Data 7), and a majority showed higher expression in the lungs than other infected tissues (Supplementary Fig. 6). The following in vitro phenotypes were most enriched in *C. neoformans* pathogenicity: membrane integrity (26/31; 84%), DNA damage response (20/31; 65%), melanin production (18/31; 58%), cell wall integrity (13/31; 42%), and growth at 37 °C (14/31; 45%) (Fig. 2). In contrast, the *sdp101Δ* mutant did not exhibit an in vitro phenotype. Notably, deletion of Sdp102, a dual-specificity MAPK phosphatase paralogous to Sdp101, resulted in minor phenotypic alteration and reduced infectivity (Figs. 2 and 3b), suggesting that Sdp101 and Sdp102 may play redundant roles in the pathogenicity of *C. neoformans*. To identify the functional correlation between Sdp101 and Sdp102, we tried to construct the *sdp101Δ sdp102Δ* double mutant strains but failed, even after repeated attempts (data not shown). Similarly, although Inp5201 is paralogous to Inp5202, we failed to generate *inp5201Δ inp5202Δ* double mutants (data not shown), likely because the *inp5201Δ* mutant alone showed severe growth defects even at 30 °C (Figs. 2 and 4a). Therefore, Sdp101/Sp102 and Inp5201/Inp5202 may have a synthetic lethal relationship. Conversely, although Oca1 and Oca101 are also paralogous, we successfully obtained an *oca1Δ oca101Δ* double mutant, but we did not find any additive or synergistic phenotypic traits in the *oca1Δ oca101Δ* double mutant compared to either single mutant (Supplementary Fig. 7), suggesting that Oca1 and Oca101 may independently contribute to the pathogenicity of *C. neoformans*.

Of the 31 pathogenicity-related phosphatases identified, five do not have evident orthologues in humans (Hs in Fig. 3c): Tps2, Siw14, Had1, Oca101, and Oca1. Therefore, these five pathogenicity-related phosphatases could be excellent anti-cryptococcal targets. Tps2, Had1, and Oca1 are also required for the virulence of *C. albicans*[25–27], thus drugs that target these phosphatases could have broad antifungal activity.

**Phosphatases involved in growth at mammalian body temperature.** We next focused on the pathobiological functions of the 31 pathogenicity-related phosphatases in *C. neoformans*. First,

because thermotolerance for mammalian body temperatures is a critical virulence factor for most human fungal pathogens, we quantitatively measured growth of each mutant at 30 and 37 °C. The *gua1Δ*, *yvh1Δ*, *fbp26Δ*, *siw14Δ*, *dbr1Δ*, *ccr4Δ*, *ppg1Δ*, *nem1Δ*, and *inp5201Δ* mutants showed impaired growth at both 30 and 37 °C (Fig. 4a and Supplementary Fig. 8). Of these, the *ccr4Δ*, *ppg1Δ*, *nem1Δ*, *dbr1Δ*, and *inp5201Δ* mutants exhibited more growth defects at 37 °C than 30 °C. The *ssu72Δ*, *phs1Δ*, *mre11Δ*, *tps2Δ*, and *cna1Δ* mutants showed impaired growth at 37 °C but not at 30 °C. A total of 14 phosphatase mutants showed impaired growth at 37 °C relative to WT (Fig. 4a and Supplementary Fig. 8) and showed reduced murine infectivity or insect virulence relative to WT (Fig. 3a and b). The *ppg1Δ*, *cna1Δ*, and *tps2Δ* mutants exhibited the most significant growth defects at 37 °C and did not grow to the level of the WT even after an extended incubation period (Fig. 4a). Concordantly, *ppg1Δ*, *cna1Δ*, and *tps2Δ* mutants exhibited highly reduced lung and brain STM values ($< -5$; Fig. 3b). The *oca101Δ* mutant showed impaired growth relative to WT at 30 °C but not at 37 °C (Supplementary Fig. 8) suggesting that the role of Oca101 in *C. neoformans* pathogenicity is not related to temperature. Collectively, these data suggest that growth at 37 °C is a critical for virulence of *C. neoformans*.

**Phosphatases involved in melanin and capsule production.** *C. neoformans* has two major virulence factors: the polyphenol pigment melanin and the polysaccharide capsule, both of which contribute to its antiphagocytic activity[28,29]. The melanin pigment also serves as antioxidant due to its reactive oxygen species scavenging activity[28]. Among 19 phosphatase mutants defective in melanin production on Niger seed medium, the following 13 mutants also exhibited defective melanin production on L-DOPA and epinephrine media (Fig. 4b): *mre11Δ*, *ccr4Δ*, *vps29Δ*, *yvh1Δ*, *fbp26Δ*, *inp5201Δ*, *cac1Δ*, *ptp2Δ*, *ptc2Δ*, *dbr1Δ*, *ppg1Δ*, *nem1Δ*, and *gua1Δ*. All of these mutants, except the *ptc2Δ* mutant, exhibited reduced infectivity or virulence (Fig. 3a and b), suggesting that melanin production is strongly correlated with pathogenicity. In contrast, *SIW14* deletion increased melanin production in all melanin-inducing media (Fig. 4b) but attenuated the virulence of *C. neoformans* (Fig. 3b), suggesting that other cellular functions mediated by Siw14 may promote virulence.

We next addressed whether these phosphatases are directly involved in induction of *LAC1*, which encodes laccase, a rate-limiting enzyme for melanin production in *C. neoformans*, under nutrient-starvation conditions[30] (Fig. 4c). Deletion of *PTP2*, *CCR4*, *INP5201*, *CAC1*, *DBR1*, *FBP26*, *GUA1*, *NEM1*, and *PPG1* significantly reduced *LAC1* induction upon nutrient starvation (Fig. 4c), with deletion of *PTP2*, *CCR4*, *CAC1*, and *PPG1* almost completely abolishing *LAC1* induction. Ptp2 has been reported as

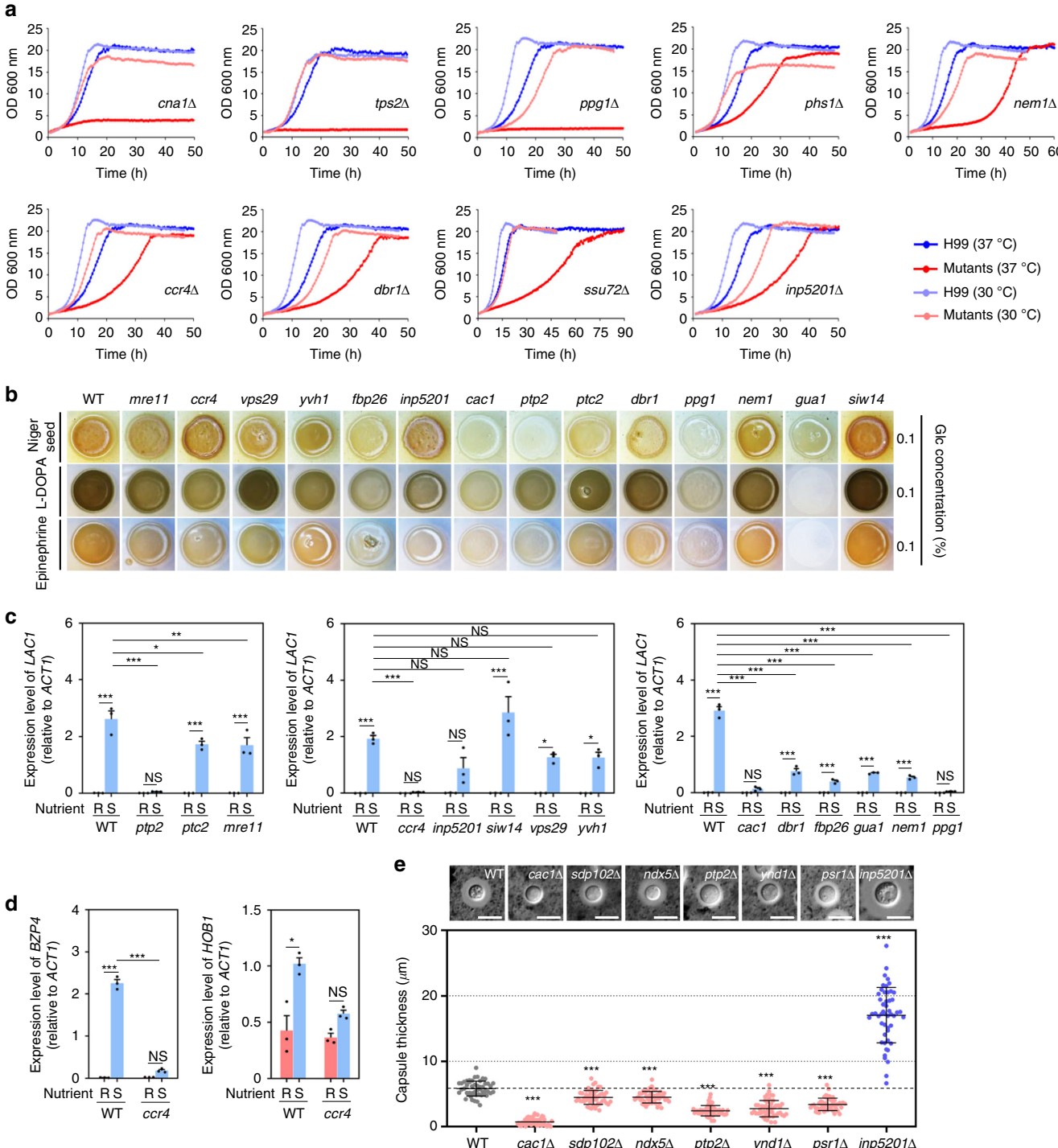

a negative feedback regulator of the Hog1 MAPK; Hog1 deletion increases *LAC1* induction and melanin production[10,30]. Recently, we have reported that *LAC1* induction is controlled by the following four core TFs: Bzp4, Usv101, Hob1, and Mbs1[30]. Among these, expression of *BZP4* and *HOB1* is induced by nutrient starvation, and *BZP4* induction itself is governed by Hob1[30]. Therefore, we examined whether the *LAC1*-regulating phosphatases also regulate induction of *BZP4* and *HOB1* (Fig. 4d and Supplementary Fig. 9). Although *BZP4* induction by nutrient starvation was significantly reduced in most melanin-defective phosphatase mutants, *CCR4* deletion in particular abolished *BZP4* (Fig. 4d) and, notably, *HOB1* induction upon nutrient starvation (Fig. 4d). Therefore, the Ccr4-Hob1-Bzp4-dependent signalling

pathway appears to be critical for *LAC1* induction and melanin production in *C. neoformans*.

We next focused on phosphatases involved in capsule production. When capsule production was quantitatively measured as packed cell volume, Cac1, Ptp2, Psr1, Ndx5, Hpp2, Oca1, Sdp102, Ynd1, Ngl3, Ppp5, and Cwh43 positively impacted cell volume, and Inp5201, Dbr1, Phs1, Cdc1, Cna1, Sit4, Ccr4, Nem1, Pcd102, and Fig4 negatively impacted it (Supplementary Fig. 10). For these mutants, we examined actual capsule thickness; among these, six mutants (*cac1Δ*, *sdp102Δ*, *ndx5Δ*, *ptp2Δ*, *ynd1Δ*, and *psr1Δ*) showed reduced capsule production (Fig. 4e and Supplementary Fig. 10) that mirrored their phenotype in cell volume assays, and five of these (*cac1Δ*, *ptp2Δ*, *sdp102Δ*, *ynd1Δ*,

**Fig. 4 Phosphatases involved in major virulence traits of *C. neoformans*. a** Growth curves of WT and phosphatase mutants were generated at 30 °C (control) and 37 °C (mammalian body temperature). Fifteen phosphatase mutants had growth defects at 30 and 37 °C. Nine of these phosphatase mutants had more substantial growth defects at 37 °C than 30 °C (additional data in Supplementary Fig. 8). Each curve represents data from two independent experiments (see Supplementary Fig. 8 for data from the individual experiments). Optical density at 600 nm (OD$_{600nm}$) was measured with a multi-channel bioreactor (Biosan Laboratories, Inc., Warren, MI, USA) for 40–90 h based on growth rate. **b** Melanin production was measured using three different melanin-inducing media (Niger seed, dopamine, and epinephrine medium). Representative images from three independent experiments are shown here. Each strain was spotted on medium containing 0.1% glucose, incubated at 30 °C, and photographed after 1–3 days. **c, d** Gene expression of the melanin-regulating genes *LAC1*, *BZP4*, and *HOB1* were determined by qRT-PCR in both nutrient-rich (R) and nutrient-starvation (S) conditions. RNA was extracted from three biological replicates with three technical replicates of WT and melanin-regulating phosphatase mutants. Expression was normalized to *ACT1*, and statistical significance was calculated by one-way ANOVA analysis with Bonferroni's multiple comparison test. Data are presented as mean values ± SEM (*$P < 0.05$, **$P < 0.001$, ***$P < 0.0001$). **e** The capsule production assay was performed using capsule-inducing media (FBS agar medium). Capsule thickness (total diameter−cell body diameter) was measured for WT cells ($n = 50$) and for each phosphatase mutant ($n = 50$). Statistical significance was calculated by one-way ANOVA analysis with Bonferroni's multiple comparison test. Data are presented as mean values ± SEM (*$P < 0.05$, **$P < 0.001$, ***$P < 0.0001$). The graph is representative of more than three independent experiments. The images are representative DIC images of WT and phosphatase mutants incubated on FBS agar medium and stained with India ink. Scale bars, 10 μm.

*psr1Δ*) also showed reduced virulence (Fig. 3). In contrast, the *inp5201Δ* mutant again showed highly enhanced capsule production but exhibited dramatically reduced virulence (STM < −7), the latter of which likely resulted from defective melanin production (Fig. 4b). Overall, the ability to produce melanin and the polysaccharide capsule were highly correlated with pathogenic potential in *C. neoformans*.

**The retromer complex promotes *C. neoformans* virulence.** Among the newly identified virulence-related phosphatases in this study, Vps29 (CNAG_00182) is a putative component of the retromer complex, first discovered in *S. cerevisiae*[31]. The yeast retromer is a cytosolic, heteropentameric protein complex that mediates the intracellular trafficking of protein cargo from the post-Golgi organelles to the lytic compartment[32,33]. It consists of the cargo-recognition core (CRC) complex, which contains Vps29, Vps35, and Vps26, and a membrane-deforming sorting nexin (SNX) complex, which contains Vps5 and Vps17[32].

To address whether functions of Vps29 resulted from the conserved role of the retromer complex, we functionally characterized other retromer components in *C. neoformans*. In the *C. neoformans* genome, we identified all of the remaining retromer component genes encoding proteins orthologous to Vps35 (CNAG_01837), Vps26 (CNAG_01426), Vps5 (CNAG_01315), and Vps17 (CNAG_00508), suggesting that the retromer complex is evolutionarily conserved in this fungal pathogen. We deleted each gene in the H99 strain and performed phenotypic analyses (Supplementary Figs. 11 and 12). Notably, deletion of *VPS35* and *VPS26* resulted in more dramatic phenotypic changes than that of *VPS29* (Fig. 5a), indicating that Vps35 and Vps26 are critical CRC complex components in *C. neoformans*. In contrast, deletion of SNX components *VPS5* and *VPS17* resulted in relatively minor phenotypic changes (Fig. 5a). Supporting this finding, CRC complex mutants showed significantly reduced virulence in the insect-killing assay whereas SNX complex mutants remained as virulent as the wild-type strain (Fig. 5b). However, both CRC and SNX complex mutants exhibited markedly reduced lung and brain infectivity in a murine-based STM analysis (Fig. 5c). Collectively, these findings indicate that the retromer CRC and SNX complexes are critical for the pathogenicity of *C. neoformans*.

**Gda1 and Ynd1 modulate *O*-mannosylation in *C. neoformans*.** In *S. cerevisiae*, Gda1 and Ynd1 are Golgi membrane-bound apyrases that modulate mannosylation of *O*-linked and *N*-linked glycoproteins and glycosphingolipids by affecting the antiport exchange ratio between GDP-mannose and GMP[34,35]. Gda1

exhibits activity highly specific to GDP whereas Ynd1 shows a much broader spectrum of activity[35,36]. In accord, double deletion of *GDA1* and *YND1* results in more severe glycosylation defects than deletion of either gene alone[35]. To investigate whether *C. neoformans gda1Δ* and *ynd1Δ* mutants show defects in *O*-glycosylation, as observed in *S. cerevisiae* and *C. albicans* mutant strains[35,37], we compared the *O*-glycan profiles of wild-type, *gda1Δ*, and *ynd1Δ* strains. The *O*-linked oligosaccharides assembled on cell wall mannoproteins (cwMPs) of *C. neoformans* cells were obtained by hydrazinolysis, labelled with 2-aminobenzoic acid (2-AA), and then analysed using HPLC with fluorescence detection (Fig. 6a). Consistent with a previous report[38], *O*-linked oligosaccharides from the *C. neoformans* wild-type strain were mostly composed of 2–4 mannose residues (Man2–Man4; M2–M4), with minor *O*-glycan species containing xylose (X1M2–X1M4) and a minor α1,2-mannotriose (M3*) species (Fig. 6a). In the *O*-glycan profile of the *gda1Δ* mutants, the M4 peak was dramatically reduced, and the M1 peak was increased compared to those in the wild-type strain (Fig. 6b). Notably, the minor xylose-containing species became more detectable in the *O*-glycans of the *gda1Δ* strain (Fig. 6b). The *ynd1Δ* mutants also showed a significantly decreased M4 peak with the increased M1 peak, although the M4 peak decreased to a lesser degree than that of the *gda1Δ* mutants (Fig. 6b). These altered profiles strongly indicate a severe defect in the elongation process of *O*-glycans, which is likely due to inefficient GDP-mannose supply from the cytosol to the lumen of the Golgi caused by decreased GDPase activity in the *gda1Δ* and *ynd1Δ* strains (Fig. 6c). Thus, the results collectively suggest that *GDA1* and *YND1* in *C. neoformans* encode membrane-bound apyrases required for Golgi *N*- and *O*-glycosylation (Fig. 6c), consistent with these genes' function in other yeasts. The more dramatic reduction of major *O*-glycans in *gda1Δ* than *ynd1Δ* indicates that Gda1p is a major GDPase responsible for GDP-mannose supply to the major *O*-glycan biosynthesis pathway. The reduced capsule size in *ynd1Δ* (Fig. 4e) suggests that Ynd1p might be also involved in the GDP-mannose supply to capsule biosynthesis. Notably, however, we failed to obtain *gda1Δ ynd1Δ* double mutants (data not shown), indicating that the two proteins may have a synthetic lethal relationship in *C. neoformans*. Because both *gda1Δ* and *ynd1Δ* mutants exhibited reduced virulence in the insect killing assay (Fig. 3a), these results collectively suggest that *O*-mannosylation is critical for virulence of *C. neoformans*.

**Phosphatases involved in the blood–brain barrier (BBB) crossing.** The most lethal damage conferred by *C. neoformans* is brain infection, which generally results in fatal meningoencephalitis. Notably, the brain STM scores for all of the phosphatase mutants

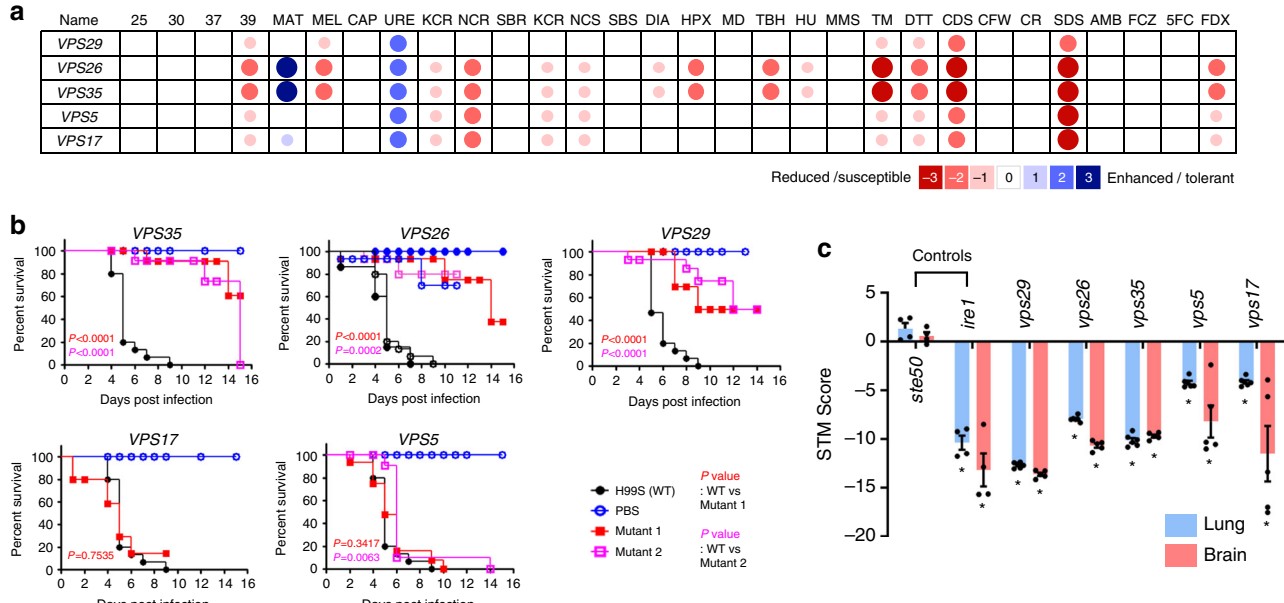

**Fig. 5 The role of retromer complex in the pathogenicity of *C. neoformans*. a** Phenotypic heatmap of retromer complex mutants constructed based on in vitro phenotypic traits [*vps29Δ* (YSB4881, YSB4882), *vps26Δ* (YSB5671, YSB5672), *vps35Δ* (YSB5615, YSB5616), *vps5Δ* (YSB5683, YSB5684), and *vps17Δ* (YSB5724)] (Supplementary Fig. 12). **b, c** The retromer complex is involved in the pathogenicity of *C. neoformans*. **b** Insect-killing assay performed for the retromer complex mutants (*n* ≥ 15). *P* values shown in the graph were calculated using the log-rank (Mantel–Cox) test to measure statistical differences between the wild-type (WT) strain (H99S) and each phosphatase mutant strain. In the *VPS26* graph, filled and empty circles indicate two independent PBS and WT-infection controls. **c** Signature-tagged mutagenesis (STM)-based murine infectivity assay (*n* = 3). STM scores were calculated by quantitative PCR. The *ste50Δ* and *ireΔ* mutants were used as virulent-positive control and avirulent-negative control strains, respectively. The statistically significant was calculated by one-way ANOVA analysis with Bonferroni's multiple comparison test. Data are presented as mean values ± SEM (*\*P* < 0.05).

were generally similar to their lung STM scores, although some of them exhibited lower brain scores (Fig. 3b; *TPS2, YMR1, FBP26, MRE11, CDC1*, and *XPP1*). However, because the phosphatase mutants administered through intranasal inhalation pass through the lungs first, those reaching the brain were less equally distributed than in pooled input mutants, which made direct comparison of lung and brain STM scores difficult. Therefore, to further address the role of these phosphatases in brain infection, we monitored the ability of the pathogenicity-related phosphatase mutants, except those showing reduced growth at 37 °C, to traverse the BBB. Among these, five mutants (*xpp1Δ, ssu72Δ, siw14Δ, sit4Δ*, and *gda1Δ*) showed significantly reduced ability to traverse the BBB (Fig. 7a). We recently showed that adhesion to the BBB is a pre-requisite for effective BBB crossing[39], and, indeed, *xpp1Δ, ssu72Δ, siw14Δ*, and *sit4Δ* mutants, but not *gda1Δ*, showed reduced BBB adhesion (Fig. 7b), suggesting that reduced *O*-mannosylation of proteins or lipids is required for BBB crossing but not adhesion. We found that *C. neoformans* does not actively grow at 37 °C in the tissue culture medium that we used for the 24 h in vitro BBB crossing and adhesion assays (data not shown), probably due to the low glucose concentration (0.1% glucose). Therefore, it is unlikely that *Cryptococcus* replication in the bottom well complicated our data.

We recently reported several TFs that promote BBB adhesion that are highly induced in vitro by host-mimic conditions (HMC; RPMI medium containing 10% FBS at 37 °C under 5% $CO_2$): *PDR802, FZC31*, and *GAT201*[39]. Therefore, we next addressed whether these genes, in addition to known brain infection-related genes [inositol transporter genes (*ITR1a* and *ITR3c*) and *MPR1*][40,41], were induced by HMC in *xpp1Δ, ssu72Δ, siw14Δ*, and *sit4Δ* mutants. Among these, *SSU72* deletion most markedly reduced HMC-mediated *MPR1* induction (Fig. 7c), suggesting that the defects of *ssu72Δ* mutants in BBB crossing and adhesion are at

least partially caused by reduced *MPR1* induction. However, because the *ssu72Δ* mutants were more defective than the *mpr1Δ* mutants in BBB crossing and adhesion (Fig. 7a and b), other cellular functions of Ssu72 may also be involved in BBB crossing and adhesion. Deletion of *SIT4* and *GDA1* markedly reduced HMC-mediated *ITR3c* induction (Fig. 7c). *GDA1* deletion also affected *FZC31* induction (Fig. 7c). However, induction of *GAT201* and *PDR802* was not significantly affected by deletion of *SIT4, SIW14, SSU72, GDA1*, or *XPP1* (Fig. 7c).

To obtain insight into the integrated signalling networks governing brain infection by *C. neoformans*, we generated functional gene networks using STRING analysis by combining data from this and previously published studies on BBB-crossing-related phosphatases, TFs, and kinases (Fig. 7d)[39]. We found that genes involved in glucose sensing (Sit4-Snf1-Gal83), RNA processing (Ssu72), and purine metabolism (Xpp1-Met3) were critical for BBB crossing by *C. neoformans*. Collectively, these findings indicate that *C. neoformans* utilizes complex signalling networks for brain infection.

**Comparison of fungal pathogenicity-related phosphatases.** By comparing the virulence data of phosphatase mutants in *C. neoformans* with those available in other plant and animal pathogenic fungi[8,25–27,42–65], we found several core fungal pathogenicity-related phosphatases (Fig. 8 and Supplementary Data 9). In the human yeast pathogens *C. neoformans* and *C. albicans*, the following 13 phosphatases are considered core pathogenicity-related phosphatases: Cna1/Cmp1, Sit4, Oca1, Yvh1, Sdp101/Cpp1, Ptp2/Ptp3, Cac1/Cyr1, Ccr4, Had1/Rhr2, Tps2, Inp5201/Inp51, Ppg1, and Gua1. Of the 13 phosphatases, CnaA, SitA, and OrlA (a Tps2 ortholog) have been shown to be required for *A. fumigatus* virulence[53–55]. Upon comparison with pathogenicity-related phosphatases in *F. graminearum*, eight

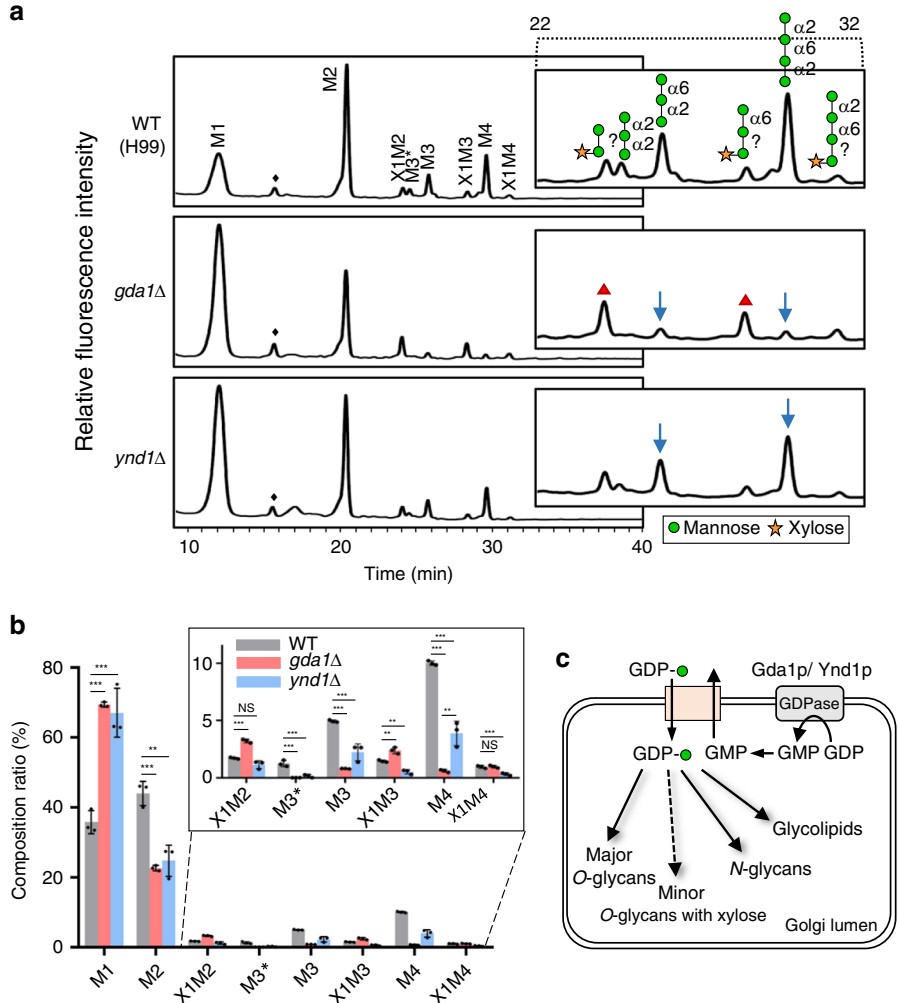

**Fig. 6 O-linked glycan profiles of *C. neoformans gda1Δ* and *ynd1Δ* strains. a** HPLC profiles of *O*-linked glycans assembled on the cell wall mannoproteins of *C. neoformans* wild-type (WT), *gda1Δ* (YSB4750), and *ynd1Δ* (YSB4856) strains. To distinguish small peaks more clearly, HPLC profiles from 22 to 32 min in the *x*- and *y*-axes were enlarged and inserted. diamond, unknown peak. M mannose, X xylose, M3* α1,2-mannotriose. Arrow, decreased M3 and M4 peak; triangle, increased *O*-glycan species containing xylose (X1M2–X1M4) in the mutant strains. **b** *O*-glycan composition of each *C. neoformans* strain. For quantitative analysis, the area of each peak in the HPLC profile from three independent experiments was measured and presented as percentage of the total area of all peaks. Statistical significance was calculated by one-way ANOVA analysis with Bonferroni's multiple comparison test. Data are presented as mean values ± standard deviation (SD) (*$P < 0.05$; **$P < 0.001$; ***$P < 0.0001$). **c** The proposed function of Gda1p and Ynd1p to provide donor substrates for mannosylation of proteins and lipids in the Golgi apparatus of *C. neoformans*.

phosphatases have been shown to be required for the virulence of both animal and plant fungal pathogens: Sit4, Yvh1, Sdp2/Msg5 (an Sdp101 orthologue), Ptp2, Ac1 (a Cac1 orthologue), Tps2, Inp53 (an Inp5201 orthologue), and Ppg1. Sit4 is involved in the TOR pathway, Cac1 is involved in the cAMP pathway, and Ppg1 and Yvh1[51,66–68] are involved in cell growth, nutrient sensing, and the stress response in fungal pathogens[2,69]. The Cna1 and Had1-mediated calcineurin pathway, the Tps2-mediated trehalose pathway, and the Msg5-mediated Mpk1/Slt2 MAPK pathway are all required for maintaining cell wall integrity. Ptp2, which is a major negative feedback regulator of the HOG pathway, is involved in adaptation and the stress response in fungal pathogens. Inp51, Inp52, and Inp53 are involved in phosphoinositide signalling, which controls vesicle trafficking, the actin cytoskeleton, and cell wall integrity[70,71]. Based on these data, phosphatases and signalling pathways involved in cell growth, nutrient sensing, cell wall integrity, the stress response, and phosphoinositide signalling appear to play pivotal roles in general fungal pathogenicity. Notably, however, deletion of *PPH3* reduces *C. neoformans*

and *F. graminearum* virulence but enhances *C. albicans* virulence[8,42]. In contrast, deletion of *PTC2* and *PTC3* reduces *C. albicans* and *F. graminearum* virulence, respectively, but does not reduce *C. neoformans* virulence[27,72]. Thus, some phosphatases may play differential roles in controlling the virulence of various fungal pathogens.

## Discussion

In this study, we identified a total of 139 phosphatases in *C. neoformans*, including protein phosphatases, lipid/nucleoside/carbohydrate phosphatases, and pyrophosphatases, and constructed a high-quality library of 219 signature-tagged gene-deletion mutant strains representing 109 phosphatases for functional analysis. By incorporating 11 additional signature-tagged mutants representing Six phosphatases that we previously constructed[4,5,10,11], we systematically analysed the in vitro and in vivo phenotypic traits of a total of 230 signature-tagged mutant strains representing 114 phosphatases (82% of original 139). Under 30 distinct in vitro growth conditions (e.g., temperature-dependent growth, mating,

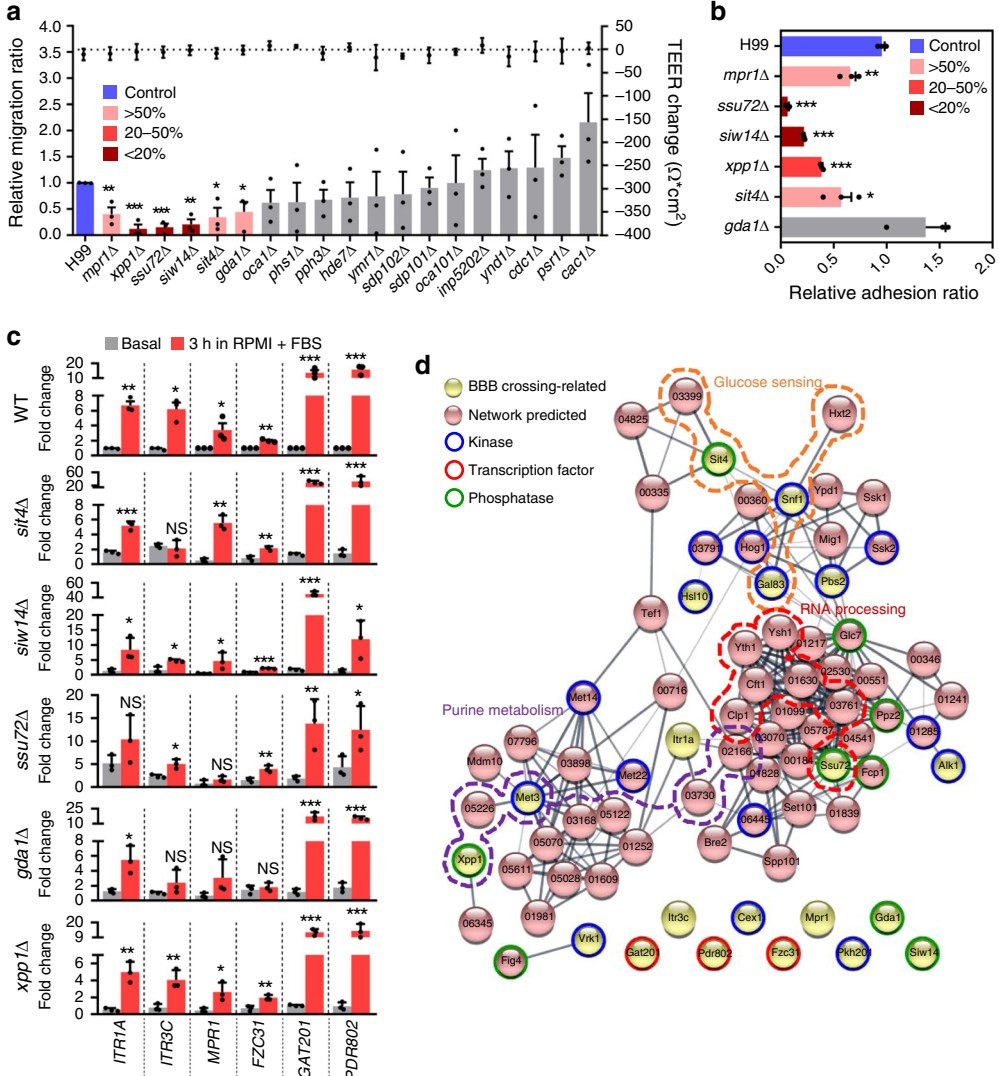

**Fig. 7 In vitro BBB transmigration and adhesion assays for *C. neoformans* phosphatases. a** In vitro BBB migration and **b** human brain microvascular endothelial cell line (hCMEC/D3) adhesion assay of pathogenicity-related phosphatases. Right: *y*-axis indicates trans-endothelial electrical resistance (TEER). Data plots represent individual data from three independent experiments (*n* = 3). Data presented as mean values ± SEM. Significant differences were calculated by two-tailed (unpaired) Student's *t*-test comparing wild-type (WT) and each phosphatase deletion mutant. **c** Host-mimic condition (HMC)-mediated induction of brain infection-related genes in wild-type (WT) and phosphatase deletion mutant strains. Gene expression was determined by quantitative RT-PCR with cDNA synthesized from total RNA prepared from cells shifted from basal condition (YPD at 30 °C, grey shaded) to HMC (RPMI with 10% foetal bovine serum at 37 °C under 5% $CO_2$ and further incubated for 3 h (red shaded). Fold-change of gene expression was calculated relative to basal expression levels of each gene in WT. Data from three independent experiments (black dots) are presented as mean values ± SEM. Statistical significance between basal and HMC was determined by two-tailed (unpaired) Student's *t*-test (*$P < 0.05$; **$P < 0.001$; ***$P < 0.0001$). **d** Functional protein association network analysis of BBB crossing-related TFs, kinases, and phosphatases of *C. neoformans* predicted by STRING (http://string-db.org) with protein sequences obtained from FungiDB (https://fungidb.org/fungidb/). Images were drawn with Cytoscape v3.7.2. Dotted lines: groups of genes involved in purine metabolism (purple), RNA processing (red), and glucose sensing (orange).

capsule/melanin/urease production, stress and antifungal drug resistance), 60 out of 114 mutants (53%) exhibited at least one phenotypic trait. Among the remaining 54 mutants that did not show a phenotypic trait under these conditions, none possessed infectivity and virulence defects, suggesting that these 54 phosphatases may not be involved in the pathobiology of *C. neoformans*. Among these, however, functionally redundant phosphatases could exist, and evident phenotypes may only result from their combined mutation. In particular, our in vivo expression profiling revealed that the following phosphatase genes were markedly upregulated during host infection: *CDC1*, *GDA1*, *GEP4*, *NDX2*, *NDX3*, *PTP3*, *APH4*, *HAD9*, *YMR1*, *INM1*, *PPU1*, *DPP101*,

and *AKP1* (Supplementary Data 8). Therefore, we recommend that the role of these phosphatases be further characterized in future work.

To identify functional correlations between TFs, kinases, and phosphatases, we attempted to make co-phenotypic clusters of TF, kinase, and phosphatase mutants constructed by this and previous studies[4,5], but found that even well-established correlations between known signalling components, such as Hog1 and Ptp2 and Cna1 and Crz1, were not evident in the co-clustering analysis. There could be several explanations for this. Mutation of a phosphatase gene, which functions as a negative feedback regulator of a kinase-dependent pathway, may not lead to a

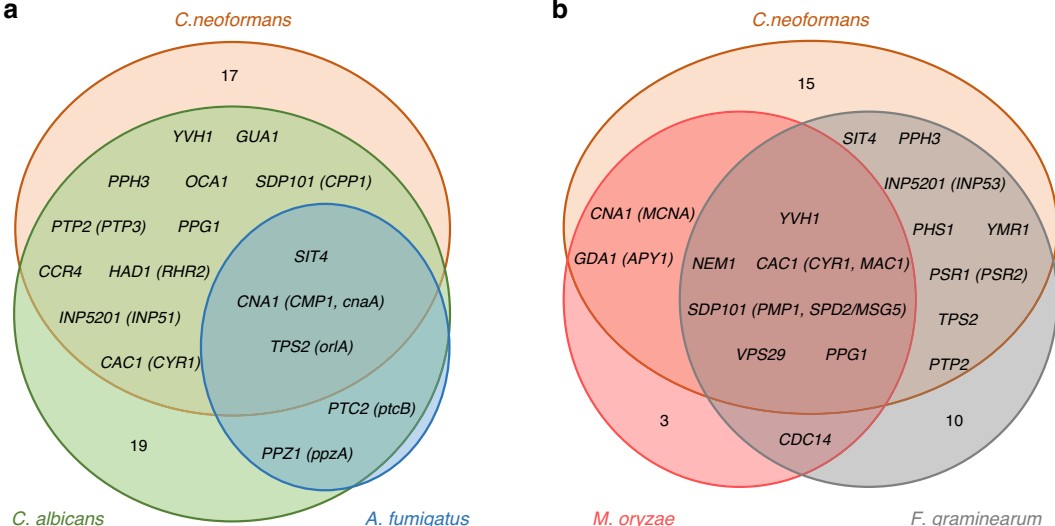

**Fig. 8 Pathogenicity-related phosphatases in fungal pathogens. a** Venn diagram showing the distribution of pathogenicity-related phosphatases in the human fungal pathogens *C. neoformans*, *Candida albicans*, and *Aspergillus fumigatus*. **b** Venn diagram showing the distribution of pathogenicity-related phosphatases in *C. neoformans* and plant-pathogenic filamentous fungi *Fusarium graminearum* and *Magnaporthe oryzae*.

phenotype opposite to the kinase mutant phenotype. For example, mutations in *PTP2*, which encodes a major negative feedback regulator of Hog1 MAPK, results in phenotypes similar to *hog1Δ* mutant phenotypes[10,73]. In addition, TF mutants generally have milder phenotypes than their upstream kinase and phosphatase mutants. For example, mutations in *CRZ1*, which encodes a downstream TF activated by Cna1 phosphatase, result in much milder phenotypes than mutations in *CNA1*[74]. Thus, simple co-phenotypic clustering of TFs, kinases, and phosphatases may lead to misinterpretations of the correlations between the signalling components. Functional and mechanistic relationships between signal components should be further investigated by RNA-seq, ChIP-seq, phosphoproteomics, and protein–protein interaction assays in future works.

Our systematic analysis unveiled both evolutionarily conserved and distinct sets of phosphatases in *C. neoformans*. Among 25 phosphatase genes that we were not able to disrupt, 14 genes (*CDC25*, *SAC1*, *FCP1*, *IPC1*, *CET1*, *GLC7*, *TIM50*, *IDI1*, *IPP1*, *RPP1*, *HIS2*, *MET22*, *PAH1*, and *GPI13*) are known to be essential in *S. cerevisiae*, *S. pombe*, or *C. albicans*: (Supplementary Data 6). The following nine genes (*CDC25*, *FCP1*, *IPC1*, *CET1*, *TIM50*, *IDI1*, *IPP1*, *RPP1*, and *GPI13*) are likely to be core essential fungal phosphatases because they are reported to be essential in *S. cerevisiae* and *S. pombe*. Out of 15 reported essential phosphatase genes in *S. cerevisiae*, we successfully deleted *PHS1*, *CYR1* (Cac1 ortholog), *CDC1*, *CDC14*, *SSU72*, and *DUT101* in *C. neoformans*. *PHS1*, *CDC1*, and *DUT101* are also essential in *S. pombe*, suggesting that they are functionally divergent between ascomycete and basidiomycete fungi or have a synthetic lethal relationship with other functionally redundant phosphatases in *C. neoformans*. We have described similar findings in our previous analysis of kinase mutant collections in *C. neoformans*[5]. Therefore, it seems evident that some kinase/phosphatase-mediated signalling networks are functionally divergent among fungi.

Here, we unravelled 31 pathogenicity-related phosphatases that affect a variety of biological processes, including growth, virulence factor production, stress response, carbohydrate metabolism, cell signalling, protein sorting, and vesicular trafficking. Some of these findings are corroborated by previous research, including the role of *PTP2*, *CAC1*, *CNA1*, *CCR4*, and *HAD1* in the virulence of *C. neoformans*[10,11,13,15,16]. Furthermore, in examining phosphatase

mutants in a large deletion mutant analysis[24], Liu et al. demonstrated that 4 of 14 mutants (*cac1Δ*, *ptc2Δ*, *sdp102Δ*, and *mre11Δ*) show reduced infectivity in the lungs[24], consistent with our findings in *cac1Δ*, *sdp102Δ*, and *mre11Δ* mutants. Notably, Madhani's and our studies indicate that the *yvh1Δ* mutant is as infective as the wild-type strain as quantified by lung STM analysis, but, in our study, the *yvh1Δ* mutant exhibited attenuated virulence in the insect-killing model. Another deviation from prior research is the secreted acid phosphatase *APH1*, which was reported by Lev et al. to contribute to the virulence of *C. neoformans*[18], but we found that deletion of *APH1* did not significantly contribute to virulence in either the insect-killing assay or murine infection model. We attribute this discrepancy to the fact that Lev et al. incubated infected *G. mellonella* at 30 °C; because when we incubated insect hosts at 37 °C, the impact of thermotolerance at mammalian temperatures may have offset any advantages seen at lower temperatures[18]. For the remaining 23 phosphatases reported here, we are the first to describe their role in the pathogenicity of *C. neoformans*.

In this study, we found that a majority of the 31 pathogenicity-related phosphatases were involved in regulation of three major cryptococcal virulence factors: thermotolerance, capsule, and melanin. Fourteen of them (Cna1, Gua1, Yvh1, Phs1, Nem1, Ppg1, Siw14, Inp5201, Ccr4, Dbr1, Ssu72, Mre11, Fbp26, and Tps2) promote thermotolerance at 37 °C. In addition, 13 phosphatases (Mre11, Ccr4, Vps29, Yvh1, Fbp26, Inp5201, Cac1, Ptp2, Dbr1, Ppg1, Nem1, Gua1, and Siw14) are involved in melanin production. Finally, capsule production was altered by deletion of six phosphatases: Cac1, Sdp102, Ptp2, Ynd1, Psr1, and Inp5201. Therefore, a total of 20 phosphatases are involved in regulation of any of the three major virulence factors, and 12 of them (Ccr4, Yvh1, Fbp26, Inp5201, Cac1, Ptp2, Dbr1, Ppg1, Nem1, Gua1, Mre11, and Siw14) are involved in modulating more than two virulence factors. The remaining 11 phosphatases (Ymr1, Oca1, Cdc1, Xpp1, Sdp101, Oca101, Gda1, Pph3, Had1, Sit4, and Inp5202) were unrelated to thermotolerance or capsule/melanin synthesis. Among these, Sdp101 and Inp5202 are paralogous to Sdp102 and Inp5201, respectively, but we failed to obtain corresponding double mutants, suggesting that Sdp101/Sdp102 and Inp5201/Inp5202 may have a synthetic lethal relationship and play redundant roles in *C. neoformans*.

The pathobiological functions of Ymr1, Cdc1, Had1, and Pph3 remain unclear because mutants of these genes do not demonstrate clear phenotypic traits related to *C. neoformans* pathogenicity. However, the *ymr1Δ*, *cdc1Δ*, and *had1Δ* mutants showed increased susceptibility to cell membrane disruption, and this instability may have contributed to reduced infectivity or virulence relative to WT. The *had1Δ* mutant also exhibited increased susceptibility to cell wall-destabilizing agents, which is in good agreement with recent reports showing that Had1 is a potential target of calcineurin, which plays a role in regulating cell wall integrity[74]. Had1 was also shown to be required for virulence in a murine model of systemic cryptococcosis[16]. *YMR1* encodes a phosphatidylinositol 3-phosphate (PI3P) phosphatase involved in vacuolar protein sorting pathways that functionally overlaps with other lipid phosphatases in *S. cerevisiae*. Therefore, deletion of *YMR1* does not generate evident phenotypes in the yeast model[75,76]. Cdc1 is a putative mannose-ethanolamine phosphatase phosphodiesterase involved in both cell division cycle and GPI-anchor remodelling[77,78]. The role of Ymr1 and Cdc1 in vacuolar protein sorting and GPI remodelling, respectively, may contribute to membrane stability in *C. neoformans*. In *S. cerevisiae*, Pph3 is a catalytic subunit of the protein phosphatase PP4 complex and governs recovery from the DNA damage checkpoint[79,80]. Pph3 likely plays a similar role in *C. neoformans* because the only in vitro phenotype observed in the *pph3Δ* mutant was increased susceptibility to DNA damage by methyl methane sulfonate (MMS). Given our previous demonstration that the DNA damage response pathway is critical for the virulence of *C. neoformans*[81,82], we hypothesize the role of Pph3 in the DNA damage response likewise contributes to virulence.

We recently identified a group of *C. neoformans* TFs and kinases involved in brain infection processes, including BBB adhesion and crossing[39]. Here we found that the following pathogenicity-related phosphatases are also involved in BBB crossing of *C. neoformans*: Xpp1, Sit4, Ssu72, Siw14, and Gda1. In *S. cerevisiae*, Ppx1 (the orthologue of Xpp1) is a metal-dependent cytosolic expolyphosphatase and generates inorganic phosphate by cleaving polyphosphate that serves as a major phosphate reservoir and is required for long-term cell survival, gene regulation, cell motility, and stress responses[83]. The functions of Sit4 and Ssu72 are likely related to the target of rapamycin (TOR) pathway involved in nutrient-sensing. In *S. cerevisiae*, Sit4 is regulated by the nutrient-sensing TOR pathway[84] and controls nitrogen catabolite repression genes through the GATA TF Gln3[85]. Prior to transcriptional termination, Ssu72 dephosphorylates the C-terminal domain of the RNA polymerase II, which is degraded through the TOR-signalling pathway in response to rapamycin[86]. Accordingly, our *sit4Δ* and *ssu72Δ* mutants indeed exhibited highly increased susceptibility to rapamycin (Supplementary Fig. 13). Notably, we found that Sit4 and Ssu72 promote induction of the inositol transporter Itr3c and metalloprotease Mpr1, respectively, required for BBB crossing of *C. neoformans*[40,41]. In *S. cerevisiae*, Siw14 hydrolyses the β-phosphatate from 5-diphosphoinositol pentakisphosphate ($IP_7$), and its deletion increases $IP_7$ and decreases $IP_6$ levels[87]. Because deletion of the inositol pentakisphosphate kinase *IPK1* decreases $IP_6$ levels, abolishing the virulence of *C. neoformans*[5,88], it is likely that Siw14 could promote BBB crossing and virulence of *C. neoformans* through modulation of the inositol signalling pathway. Finally, Gda1 is a Golgi membrane-bound apyrase that modulates *O*-mannosylation in yeast[34,35]. We previously reported that the extended structure of *O*-glycans by mannose addition in the Golgi is required for cell integrity and full pathogenicity of *C. neoformans*[38]. Based on these observations, we conclude that phosphate utilization, TOR-mediated nutrient sensing, inositol-signalling, and *O*-mannosylation are critical for BBB crossing and brain infection.

The 31 pathogenicity-related phosphatases that were identified here and in previous studies could be potential antifungal drug targets, especially Sit4, Cna1, and Tps2, which are required for virulence of the major human fungal pathogens *C. neoformans*, *C. albicans*, and *A. fumigatus*. Sit4, Yvh1, Sdp101, Ptp2, Cac1, Tps2, Inp5201, and Ppg1 are required for virulence of both human and plant fungal pathogens, thus could be targets for the development of broad-spectrum antifungal drugs. Tps2 is a particularly promising antifungal drug target because the trehalose pathway is missing in humans[89]. Recently, the structure of the N-terminal domain of *C. albicans* Tps2 has been resolved[90] and provides a structural basis for the design of Tps2-specific antifungal agents. Siw14, Had1, and Oca1/101 could also be good targets for cryptococcal treatment because there are no evident orthologues of these phosphatases in humans. Recently, the structure of the *S. cerevisiae* inositol phosphatase Siw14 has also been resolved[91], whereas structural information for Had1 and Oca1/Oca101 is not yet available.

Several clinical strategies have targeted phosphatases directly by targeting the catalytic subunit of phosphatase complexes or indirectly by targeting the regulatory or scaffolding subunits of phosphatase complexes. Examples of the former strategy include LB-100 inhibition of the PP2A-C subunit[92], FK506 inhibition of the calcineurin complex[93–95], and inhibition of dual-specificity phosphatases by several small molecules[96]. An example of the latter strategy includes ceramide and its derivatives, which are bioactive sphingolipid molecules that activate PP2A by preventing SET inhibitor binding to PP2A[97,98]. To increase the specificity of a drug that targets a phosphatase, both the catalytic and regulatory subunits of the phosphatase complex should be considered, such as was considered with the drug FK506, which targets both the catalytic and regulatory subunits of the calcineurin complex via FKBP12[95]. Based on BLASTP analysis using annotated information from the *Saccharomyces* Genome Database (SGD, http://yeastgenome.org), we found 26 putative phosphatase regulatory subunit orthologues in *C. neoformans* (Supplementary Data 10). However, further research is needed to characterize the functional and mechanistic relationships between the catalytic and regulatory subunits of phosphatases; this research could lead to the development of more antifungal drugs that target phosphatases.

## Methods

**Ethics statement**. Animal care and all experiments were conducted in accordance with the ethical guidelines of the Institutional Animal Care and Use Committee (IACUC) of Yonsei University. The Yonsei University IACUC approved all vertebrate studies.

**Construction of the *C. neoformans* phosphatase and retromer mutants**. We constructed phosphatase and retromer mutant strains in the *C. neoformans* serotype A H99S strain background through homologous recombination using gene-disruption cassettes containing the nourseothricin-resistance marker (nourseothricin acetyl transferase; *NAT*) using *NAT*-split marker/double joint PCR (DJ-PCR) strategies[99] (see Supplementary Data 4 for primers). We amplified 5′- and 3′-flanking regions of target genes by PCR with primer pairs L1/ L2 and R1/R2, respectively, from H99S genomic DNA. The signature-tagged *NAT* marker was amplified by PCR with primers M13Fe (M13 forward extended) and M13Re (M13 reverse extended) from a pNAT-STM plasmid containing the *NAT* gene with each unique, signature-tagged sequence. After amplification of the 5′- and 3′-flanking regions and *NAT* marker in the first round of PCR to generate template DNA, the second round of PCR constructed 5′- and 3′-regions of the *NAT*-split gene-disruption cassette with primer pairs L1/NSL and R2/NSR, respectively. Biolistic transformation introduced the *NAT* gene-disruption cassettes by incubating the H99S strain in 50 ml YPD medium for 16 h at 30 °C, followed by spin-down, resuspension in 5 ml distilled water, application on YPD agar medium containing 1 M sorbitol, and further incubation for 3 h at 30 °C. Gene-disruption cassettes were then combined with 600 μg of 0.6-μm gold microcarrier beads (Bio-Rad Laboratories, Hercules, CA, USA) and introduced into cells using a particle delivery

system (PDS-100, Bio-Rad). After 4 h incubation at 30 °C for recovery of cell membrane integrity, cells were scraped and spread on YPD agar medium containing nourseothricin (100 μg/ml). Diagnostic PCR identified *NAT*-positive transformants. Southern blot analysis confirmed the genotype of each screened transformant, and we constructed at least two independent mutant strains for each phosphatase gene.

**nCounter in vivo gene expression profiling analysis**. Using previously reported RNA samples[39], we performed an nCounter gene expression analysis (NanoString) to quantify in vivo expression levels of the 139 phosphatases. Total RNA samples were obtained from 6-week-old female A/J mice infected with $5 \times 10^5$ cells through nasal inhalation. Each group of three mice was sacrificed 3, 7, 14, or 21 dpi, and the lungs, brain, spleen, and kidneys were recovered and lyophilized. Dried organs were homogenized, and total RNA was extracted with a commercial RNA extraction kit (easy-BLUE, Intron Biotechnology). Samples containing 10 ng of total RNA isolated from *C. neoformans* grown under in vitro basal conditions (30 °C; YPD medium) or 10 μg of total RNA isolated from *C. neoformans*-infected mouse tissues were reacted with the custom-designed probe code set according to the manufacturer's standard protocol of the nCounter multiplex platform (NanoString, Seattle, WA, USA)[22,23]. Scanning was performed by digital analyser through high resolution (600 fields) option and normalized by nSolver software (provided by NanoString). A total of eight housekeeping genes (mitochondrial protein, CNAG_00279; microtubule-binding protein, CNAG_00816; aldose reductase, CNAG_02722; cofilin, CNAG_02991; actin, CNAG_00483; tubulin β chain, CNAG_01840; tubulin α-1A chain, CNAG_03787; histone H3, CNAG_04828) were used for expression normalization[39,100]. Normalized data were transformed to $\log_2$ scores to express fold-change and subjected to clustering using one minus Pearson correlation with average linkage by Morpheus (Broad Institute, Cambridge, MA, USA, http://software.broadinstitute.org/morpheus).

**Growth and chemical susceptibility test**. To analyse the susceptibility of each phosphatase mutant to in vitro stress conditions, *C. neoformans* was grown for 16 h at 30 °C, serially diluted 10-fold ($1–10^4$), and spotted on YPD agar media containing the following chemical agents to induce environmental stress as described previously[4,5]: osmotic stress (sorbitol) and cation/salt stresses (NaCl and KCl) under either glucose-rich (YPD) or glucose-starved (yeast extract-peptone; YP) conditions; oxidative stress [hydrogen peroxide ($H_2O_2$), *tert*-butyl hydroperoxide (an organic peroxide), menadione (a superoxide anion generator), diamide (a thiol-specific oxidant)]; toxic heavy metal stress [cadmium sulfate ($CdSO_4$)]; genotoxic stress (methyl methanesulphonate and hydroxyurea); membrane destabilizing stress [sodium dodecyl sulfate (SDS)]; cell wall destabilizing stress (calcofluor white and Congo red); ER stress [tunicamycin and dithiothreitol (DTT)]; and antifungal drug susceptibility (fludioxonil, fluconazole, amphotericin B, and flucytosine). Cells were incubated at 30 °C for 1–5 days and photographed daily. To examine the growth of *C. neoformans* strains at different temperatures, we spotted serially diluted cells on YPD agar medium, incubated them at 25, 37, and 39 °C, and photographed the cultures daily. To quantitatively examine the growth rate of phosphatase mutants, the WT strain (H99S) and phosphatase mutants were incubated at 30 °C overnight and sub-cultured into fresh liquid YPD medium [optical density at 600 nm ($OD_{600}$) = 0.2]. Cells were than incubated at 30 or 37 °C in a multi-channel bioreactor (Biosan Laboratories, Inc., Warren, MI, USA), and $OD_{600nm}$ was automatically measured for 40–90 h.

**Mating assay**. To examine unilateral mating efficiency, each serotype A *MAT*α phosphatase mutant constructed in the H99S strain and *MAT***a** KN99**a** strain were cultured in YPD medium for 16 h at 30 °C, washed twice with phosphate-buffered saline (PBS), mixed at equal concentrations ($10^7$ cells/ml), spotted on V8 mating media (pH 5), and incubated at 25 °C in the dark for 7–14 days. Filamentous growth was observed and photographed weekly.

**In vitro virulence factor production assay**. To test capsule production efficiency, each mutant was cultured at 30 °C, spotted onto Dulbecco's modified Eagle's (DME) agar medium, and incubated at 37 °C for 2 days, after which cells were scraped, washed with distilled water, fixed with 10% formalin, and washed again with distilled water. Fixed cells were adjusted to a concentration of $3 \times 10^8$ cells/ml, and 50 μl of the cell suspension was injected into microhaematocrit capillary tubes (Kimble Chase, Rockwood, TN, USA). Capillary tubes were placed vertically for 10 days to pack cells by gravity. Packed cell volume ratio (packed cell phase/total phase) was measured, and the relative packed cell volume of each mutant was calculated by normalizing each ratio with packed cell volume ratio of wild-type H99S strain. Statistical differences in relative packed cell volume ratios were determined by one-way analysis of variance (ANOVA) with Bonferroni's multiple comparison test in Prism 8 (GraphPad, San Diego, USA). To measure capsule production efficiency in other media, 5 μl of each culture was spotted on Littman's agar medium[101] and FBS agar medium (10% of foetal bovine serum and 90% of PBS), incubated at 37 °C for 2 days, scraped, and resuspended with distilled water. Resuspended cells were stained by India ink (BactiDrop; Remel, San Diego, CA, USA) and observed by differential interference contrast (DIC) microscopy (BX51, Olympus, Tokyo, Japan). Capsule thickness was measured by subtracting the cell

diameter from capsule diameter (total diameter−cell body diameter). For quantitative measurement of capsule thickness, 50 cells were measured for the H99S strain and each phosphatase mutant. To examine melanin production efficiency, each phosphatase mutant was cultured for 16 h in YPD medium at 30 °C, washed with PBS, and spotted (3 μl) on Niger seed, dopamine, or epinephrine media (1 g L-asparagine, 3 g $KH_2PO_4$, 250 mg $MgSO_4$, 1 mg thiamine, 5 μg biotin, and 100 mg L-DOPA or epinephrine hydrochloride per litter) containing 0.1% or 0.2% glucose. Spotted cells were incubated at 37 °C and photographed after 1–3 days. For the phosphatase mutants growth-defective at 37 °C, melanin and capsule production efficiency were examined at 30 °C. To examine urease production, each phosphatase mutant was cultured at 30 °C for 16 h, washed with PBS, and inoculated ($10^6$ cells) onto liquid Christensen's media in a 10-ml medical tube (SPL Life Sciences, Gyeonggi, Korea), then incubated at 30 °C in a shaking incubator for 1–3 days and photographed daily.

**Expression analysis**. To measure the expression level of known melanin-regulating genes (*LAC1*, *HOB1*, *MBS1*, and *BZP4*), the H99S strain and phosphatase mutants (*mre11*Δ, *ccr4*Δ, *vps29*Δ, *yvh1*Δ, *fbp26*Δ, *inp5201*Δ, *cac1*Δ, *ptp2*Δ, *ptc2*Δ, *dbr1*Δ, *ppg1*Δ, *nem1*Δ, and *gua1*Δ) were incubated in liquid YPD medium for 16 h at 30 °C and sub-cultured to fresh liquid YPD medium ($OD_{600} = 0.2$). When the cells reached the early logarithmic phase ($OD_{600} = 0.6–0.8$), half of the cell culture was sampled to prepare a basal sample. The remaining cell culture was washed three times with PBS and incubated in nutrient-starvation conditions (YNB medium with ammonium sulfate without glucose) for 2 h, then immediately pelleted with liquid nitrogen and lyophilized. To measure the expression level of BBB crossing-related genes (*ITR1A*, *ITR3C*, *MPR1*, *FZC31*, *GAT201*, and *PDR802*), the H99S strain and phosphatase mutants [*sit4*Δ (YSB4094), *siw14*Δ (YSB4570), *ssu72*Δ (YSB4242), *gda1*Δ (YSB4750), and *xpp1*Δ (YSB5941)] were incubated in YPD broth at 30 °C, sub-cultured into 50 ml of fresh YPD broth, and further incubated until $OD_{600}$ reached 0.8. The culture was then separated into two 25-ml tubes, centrifuged, and washed three times with sterile distilled water. One tube was kept in liquid nitrogen to monitor basal expression levels, and the other tube was resuspended with an equal volume of RPMI1640 medium containing 10% FBS. After 3 h incubation at 37 °C in a $CO_2$ incubator with 120 rpm horizontal shaking, cells were centrifuged and lyophilized overnight. Total RNA was extracted from each sample using a commercial RNA extraction kit (easy-BLUE, iNtRON Biotechnology, Gyeonggi, Korea) and cDNA was synthesized using RTase (Thermo Scientific, Waltham, MA, USA). Quantitative reverse transcription-PCR (qRT-PCR) was performed with target gene-specific primer pairs listed in Supplementary Data 4.

**HPLC analysis of *O*-linked glycans from cwMPs**. Analysis of *O*-linked glycans from cwMPs was conducted as described previously[38,102]. The *O*-linked oligosaccharides were released from the purified cwMPs by modified hydrazinolysis. Dried cwMPs (50 μg) were resuspended in hydrazine monohydrate and incubated at 60 °C for 4 h. After cooling and desiccation of the reactant, the pellets were dissolved in $NaHCO_3$, mixed with $(CH_3CO)_2O$, and incubated on ice for 30 min. *O*-glycans were purified by Dowex 50WX8-400 resins (Sigma-Aldrich, St. Louis, MO, USA) and labelled with 2-AA. The purified *O*-glycan was analysed using HPLC on a TSKgel Amide-80 column (0.46 × 25 cm, Tosoh Corp., Tokyo, Japan) with 90% solvent A (2% acetic acid and 1% tetrahydrofuran in acetonitrile) and 10% solvent B (5% acetic acid, 3% triethylamine, and 1% tetrahydrofuran in water). After sample injection, the proportion of solvent B increased to 90% over 60 min at a flow rate of 1.0 ml/min and then *O*-glycans were detected with a fluorescence detector (2475, Waters Corp., Milford, MA, USA) at excitation and emission wavelengths of 360 and 425 nm, respectively. Data were analysed by using chromatography software (Empower 2, Waters).

**Insect-based in vivo virulence assay**. At least 15 *G. mellonella* caterpillars (Vanderhorst Wholesale, Inc., Saint Marys, OH, USA) in the final larval instar with a body weight of 200–300 mg, arriving within 7 days from the day of shipment, were used for in vivo insect virulence assays. Each phosphatase mutant and wild-type H99S strain was incubated at 30 °C overnight, pelleted, washed three times with PBS, and resuspended in PBS at a concentration of $10^6$ cells/ml. Four thousand *C. neoformans* cells per larvae were injected into the second-to-last prolegs with a 100-μl syringe equipped with a 10-μl needle and repeating dispenser (PB600-1, Hamilton Company, Reno, NV, USA). Negative control *G. mellonella* received PBS only. Infected larvae were placed in Petri dishes in a humidified container, incubated at 37 °C, and monitored daily. Larvae were considered dead when they turned black and showed no movement upon touching. Larvae that pupated during the experiment were censored for statistical analysis. Survival curves were illustrated using Prism 8 (GraphPad, San Diego, CA, USA) and analysed with a log-rank (Mantel–Cox) test. We examined two independent strains for each phosphatase mutant.

**STM-based murine infectivity assay**. A set of phosphatase mutants with the 41 unique signature-tagged *NAT* selection markers was cultured for 16 h at 30 °C. The *ste50*Δ (STM#282) and *ire1*Δ (STM#169) mutants were used as virulent and avirulent control strains, respectively, as described previously[103,104]. Mutants and

control strains were pelleted, washed three times with PBS, resuspended in PBS, and then pooled at equal numbers ($5 \times 10^5$ cells). Seven-week-old female A/J mice (Jackson Laboratory, Bar Harbor, ME, USA) anaesthetized with an intraperitoneal injection of Avertin (2,2,2-tribromoethanol, T48402, Sigma-Aldrich, St. Louis, MO, USA) were infected with $5 \times 10^5$ cells of pooled mutants (in 50 μl PBS) through intranasal inhalation. To prepare the input phosphatase genomic DNA library, 200 μl of pooled strains were spread on YPD media containing 100 μg/ml chloramphenicol, incubated at 30 °C for 3 days, and collected by scraping. The infected mice were sacrificed with an overdose of Avertin 14 dpi. Lungs and brains of infected mice were recovered and homogenized with 5 ml of PBS. Homogenized tissues were then spread on YPD medium containing 100 μg/ml chloramphenicol, incubated at 30 °C for 3 days, and collected by scraping. Genomic DNA was extracted from collected input and output cells using the cetyl trimethylammonium bromide (CTAB) method[103]. Quantitative PCR was performed with tag-specific primers (Supplementary Data 4) using a qRT-PCR system (CFX96, Bio-Rad, Hercules, CA, USA). The STM score was calculated using the $2^{-\Delta\Delta CT}$ method[105] to determine STM score to calculate the relative changes in genomic DNA amounts. The mean fold-changes in input verses output samples were calculated as a log score ($\log_2 2^{-(Ct,Target-Ct,Actin)output-(Ct,Target-Ct,Actin)input}$).

**In vitro BBB-crossing and BBB-adhesion assays**. Human brain microvascular endothelial cells (hCMEC/D3 cell line, Merck & Co., Kenilworth, NJ, USA) were cultured as follows based on the reported methods[39,106,107]. Briefly, $5 \times 10^4$ hCMEC/D3 cells in EGM-2 medium (Lonza Group, Basel, Switzerland) were seeded on collagen (Corning, Inc., Corning, NY, USA)-coated 8-μm porous membranes (BD Biosciences, Franklin Lakes, NJ, USA) for the BBB-crossing assay or 12-well plates (BD Biosciences) for the BBB-adhesion assay. The day after seeding, medium was refreshed with EGM-2 medium supplemented with 2.5% human serum and further incubated for 4 days. A day before *C. neoformans* inoculation, the medium was replaced with 0.5 × diluted EGM-2 medium, and the cells were maintained at 37 °C and 5% $CO_2$. The integrity of tight junctions between cells was confirmed by verifying that the trans-endothelial electrical resistance (TEER) measured ~200 Ω/$cm^2$, as assed by an epithelial volt/ohm meter (EVOM² device, World Precision Instruments, Sarasota, FL, USA).

For the BBB-crossing assay, $5 \times 10^5$ cells of *C. neoformans* WT (H99), *mpr1Δ* mutant, and phosphatase deletion mutants were added to 500 μl of PBS and inoculated onto the porous membranes. After 24 h incubation at 37 °C in 5% $CO_2$, the number of yeast cells passing through the porous membrane was measured by counting CFU. Tight junction integrity was measured by TEER as described above before and after inoculation of yeast cells. The BBB migration ratio was calculated by dividing the output CFU of each tested strain by that of WT. For the BBB-adhesion assay, $5 \times 10^5$ yeast cells in 100 μl of PBS were inoculated onto a monolayer of hCMEC/D3 cells grown in a 12-well plate and incubated for 24 h at 37 °C in 5% $CO_2$. Following incubation, cultures were washed three times with PBS, incubated with sterile distilled water for 30 min at 37 °C to burst the host cells, and collected for CFU quantification. The BBB-adhesion ratio was calculated by dividing the adhered CFU of each test strain by that of WT *C. neoformans*.

**C. neoformans phosphatase web-accessible database**. We developed the *Cryptococcus neoformans* Phosphatase Phenome Database (http://phosphatase.cryptococcus.org) and *Cryptococcus neoformans* Phenome Gateway Database (http://www.cryptococcus.org/) to facilitate public access to the phenomic and genomic data for the *C. neoformans* phosphatase mutant library. The database was built under the same software development environment as described in our two previous publications[4,5]. Genome sequences and annotations of the H99S strain were obtained from the standardized genome data warehouse in the Comparative Fungal Genomics Platform Database (CFGP 2.0; http://cfgp.riceblast.snu.ac.kr)[108]. Except reported phosphatases, those first functionally characterized by this study were named based on the published gene nomenclature rules for *C. neoformans*[109] (Fig. 1a).

**Statistical analysis**. Statistical analyses were performed with GraphPad Prism version 8. For the capsule production assay and expression analysis, ANOVA with Bonferroni's multiple comparison test was used. For the insect-killing assay, the log-rank (Mantel–Cox) test was used for statistical analysis. For the murine STM analysis, the statistical significance between *ste50Δ* (positive control) and mutants was calculated by one-way ANOVA with Bonferroni's multiple comparisons test.

**Reporting summary**. Further information on research design is available in the Nature Research Reporting Summary linked to this article.

## Data availability

The phosphatase domain and sequence data in Fig. 1 were retrieved from FungiDB (https://fungidb.org/fungidb/) and InterPro (https://www.ebi.ac.uk/interpro/). All data to classify the phosphatases in Fig. 1a and b are available in Supplementary Data 1. Detailed information about phosphatases in *S. cerevisiae*, *S. pombe*, *C. albicans* and *U. maydis* is listed in Supplementary Data 3. Information about gene IDs and names, strain numbers, and genotypes with signature tag numbers for phosphatase mutant strains used in this

study is available in Supplementary Data 5. We provide the whole phosphatase mutant collection via Fungal Genetics Stock Center (FGSC, http://www.fgsc.net/) in USA, Korean Culture Center of Microorganisms (KCCM, http://www.kccm.or.kr/) and Korean Collection for Type Cultures (KCTC, https://kctc.kribb.re.kr/) in South Korea. The whole phenome data for phosphatase mutants are available in the *Cryptococcus neoformans* Phosphatase Phenome Database (http://phosphatase.cryptococcus.org). Integrated phenome data of transcription factor, kinase, and phosphatase mutants in *C. neoformans* in this and previous studies[4,5] are available in the *Cryptococcus neoformans* Phenome Gateway Database (http://www.cryptococcus.org/), in which individual gene/protein information is linked to that of FungiDB. The whole NanoString-nCounter® analysis data for in vivo phosphatase gene expression and probe information are available in Supplementary Data 8. The source data underlying Supplementary Figs. 7b, 11b, d, f and h are provided as a Source Data file. Source data are provided with this paper.

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

## Acknowledgements

This work was supported by the Strategic Initiative for Microbiomes in Agriculture and Food funded by the Ministry of Agriculture, Food and Rural Affairs (grant 916006-2 and 918012-4 to Y.-S.B.) and, in part, by National Research Foundation of Korea grants (grants 2016R1E1A1A01943365 and 2018R1A5A1025077 to Y.-S.B.; 2018R1C1B6009031 to K.-T.L.) from the Ministry of Science and ICT, and Brain Korea 21 (BK21) PLUS programme. This work was also supported by NIH/NIAID R37 MERIT award AI39115-21 and NIH/NIAID RO1 award AI50113-15 to J.H. J.H. is co-director and fellow CIFAR programme Fungal Kingdom: Threats & Opportunities.

## Author contributions

Y.-S.B. conceived the project. J.-H.J., K.-T.L., J.H., D.L., E.-H.J., J.-Y.K., Y.L., S.-H.L., Y.-S.S., K.-W.J., D.-G.L., E.J., M.L., Y.-B.J., Y.C., M.H.L., J.-S.K., S.-R.Y., J.-T.C., J.-W.L., H.C., S.-W.K., K.J.S., Y.L., E.J.T., J.C., and A.F.A. performed experiments and analysed the data. Y.-H.L., J.H., H.A.K., E.C., and Y.-S.B. supervised the experimental analysis. J.-H.J., K.-T.L., Y.-H.L., J.H., H.A.K., E.C., and Y.-S.B. wrote the manuscript. All authors have reviewed and approved this manuscript.

## Competing interests

The authors declare no competing interests.
