## [Peer Review File · Nature Communications]

REVIEWER COMMENTS

Reviewer #1 (Remarks to the Author):

The paper at hand addresses the function of the *C. neoformans* phosphatases by analyzing the phenotypical traits and virulence of 230 mutants representing 114 phosphatases. It is a logical follow up of previous work of this group, which carried out a similar systematic functional profiling of transcription factors and kinases by using approaches akin to the ones followed in this work. This has the drawback that the novelty of this manuscript is not exceptional, but it paves the road for a very well designed, executed, analysed and presented experimental work, which extracts relevant information especially interesting for researchers working on eukaryotic signal transduction and fungal pathogenesis.

The manuscript provides very interesting data on the impact of the activity of every not essential phosphatase on fundamental aspects of the biology of this fungus, especially on the sensitivity to distinct stresses and, more importantly, on the most relevant *C. neoformans* virulence factors, including thermotolerance, capsule and melanin production. Furthermore, the authors perform large-scale virulence assays using insect and murine model systems to evaluate the impact of the distinct phosphatases on pathogenicity. In addition, the work identifies the essential role of Vps29 and the retromer complex in virulence as well as the important role of Gda1 and Ynd1 in fungal pathogenesis through modulation of O-mannosylation. The work is well written and the data and conclusions presented will really be useful for future scientific work.

Reviewer #2 (Remarks to the Author):

In this study, the authors made 114 phosphatase mutants in *Cryptococcus neoformans* and analyzed their various *in vitro* phenotypes. They also assessed their virulence level in an insect and a mouse model. I applaud this group for their continuous effort making large strain collections (after the transcription factor and kinase deletion mutants) and make them publicly assessable. These strains and the identified phenotypes associated with each strain will be a valuable resource to the research community.

I have only a few comments/suggestions:

1. One thing that will significantly increase the value of the current work is to incorporate the current phosphatase data with previous data on TFs and kinases in the same phosphatase database. As the authors stated, the goal of this work was to find out the interplay of the phosphatase network with the TF and kinase networks. For example, if you search a certain phosphatase, you should be able to get links to kinases or TFs that have some connections to this phosphatase based on correlated phenotypes or expression profiles.
2. The *in vivo* expression of these genes in different organs was analyzed at DPI 3, 7, 14 and 21. However, in the inhalation infection model, there is no dissemination of *Cryptococcus* from lungs to other organs by day 3. Often no dissemination could be detected even at day 7. Please explain this in detail in manuscript.
3. In the *in vitro* BBB crossing assays, CFUs were counted from the bottom well after 24 hours of incubation. *Cryptococcus* replication in the bottom well will complicate the measure of fungal cells transmigrated from the top well. Similar issues exist for adhesion assays.

Reviewer #3 (Remarks to the Author):

This manuscript reports the molecular characterization of the whole set of phosphatases in the human fungal pathogen *Cryptococcus neoformans*. The authors provided a very deep analysis of the phosphatase null mutants that surely will benefit the large community working with *C. neoformans*. Most of their findings are novel and open several new questions and research

opportunities for studying signal transduction in human pathogenic fungi. That is an amazing effort to have a comprehensive understanding about the phosphatase function in this organism. The authors also try to integrate previous information (provided by their own work) about the collaboration among phosphatases, kinases, and transcription factors. This manuscript deserves publication in Nature Communications. I would like to provide few suggestions that are necessarily mandatory to allow the manuscript acceptance.

1) The authors failed to obtain any viable transformants for some of the phosphatases (or obtained only potential aneuploid mutants possessing both wild-type and mutant alleles), suggesting these genes may be essential. Is not possible to construct conditional mutants at least for some of these gene candidates aiming to validate their hypothesis about gene essentiality ?

2) there was some discussion about the influence of calcineurin in the regulation of HAD1. I noticed that there were no experiments in Figure 3 with calcineurin inhibitors like cyclosporine or FK506. Also there was no experiment with rapamycin. These drugs are important to implicate phosphatases in the function of calcineurin or TOR.

3) It could be interesting some speculation about the number of phosphatase regulatory subunits present in *C. neoformans* genome.

4) An annoying detail: Mre11 is the name for a nuclease subunit of the MRX complex with Rad50p and Xrs2p. This complex functions in repair of DNA double-strand breaks and in telomere stability. Is it not possible to rename the phosphatase Mre11 ?

Reviewer #4 (Remarks to the Author):

This manuscript by Jin et al performed systematic analyses on the phosphatase mutants in the human fungal pathogen *Cryptococcus neoformans*. This is a very comprehensive study with solid data. However, given this is the 4th -omic study from this group, the authors should have better dataset summarization and organization rather than simply listing each phenotypic score or each growth curve. Also, if the authors wanted to emphasize the potential drug target value of certain phosphatases, it is better to include the discussion of feasibility of drug targeting, for example, to compare the similarity between fungal phosphatase and human's. There are a lot of conserved essential genes across fungal species, but not suitable as drug target at all.

Specific points:

Figure 1C: Please make sure the pie charts of each species have the same size and format.

Figure 2: It's fine to display the homology level of each phosphatase, but not very informative for the central question of the study. The authors clustered the kinase mutants according to the phenotypic assays in their previous work, that's a very nice functional analysis. Can such kind of analysis be performed here?

Figure 3A: please label number of larvae each test group used. And please double check the p value of each test, for example, in the panel of GDA1, the orange curve is closer to the black curve (WT control), but the p value of orange curve is smaller than the pink one. Figure 3C: For the phenotypic traits evaluation, are these the sub-dataset from Figure 2? It is redundant data presentation if no new information provided. Also, is there any particular reason using different size of dot other than the color gradient?

Figure 4-6: It is better to group the virulence traits assays in one figure, with representative mutants.

Figure 8: The authors showed the abnormal mannosylation of GDA1 and YND1 mutation, however, they did not really establish the relationship between mannosylation and virulence in these two mutants. With current data, it is not appropriate to state "xxx modulate virulence of *C. neoformans* by controlling O-mannosylation".

Figure 9: I wonder if the in vitro BBB traversing data could correlate with in vivo results, by intravenously infect mice with the mutants and check the brain level. In vitro assay results need to be confirmed by in vivo data.

Figure 10: The idea of "core pathogenicity-related phosphatases" is not very clear. What specific pathways or functions related to these phosphatases make them broadly related to virulence (or actually in vivo fitness)? The authors should at least have more discussion on this instead of purely listing the overlapped or not overlapped genes.

[Responses to reviewers' comments]

REVIEWER COMMENTS

Reviewer #1 (Remarks to the Author):

The paper at hand addresses the function of the *C. neoformans* phosphatases by analyzing the phenotypical traits and virulence of 230 mutants representing 114 phosphatases. It is a logical follow up of previous work of this group, which carried out a similar systematic functional profiling of transcription factors and kinases by using approaches akin to the ones followed in this work. This has the drawback that the novelty of this manuscript is not exceptional, but it paves the road for a very well designed, executed, analysed and presented experimental work, which extracts relevant information especially interesting for researchers working on eukaryotic signal transduction and fungal pathogenesis.

The manuscript provides very interesting data on the impact of the activity of every not essential phosphatase on fundamental aspects of the biology this fungus, especially on the sensitivity to distinct stresses and, more importantly, on the most relevant *C. neoformans* virulence factors, including thermotolerance, capsule and melanin production. Furthermore, the authors perform large-scale virulence assays using insect and murine model systems to evaluate the impact of the distinct phosphatases on pathogenicity. In addition, the work identifies the essential role of Vps29 and the retromer complex in virulence as well as the important role of Gda1 and Ynd1 in fungal pathogenesis through modulation of O-mannosylation. The work is well written and the data and conclusions presented will really useful for future scientific work.

Response: We thank the reviewer for appreciating the quality of our work.

Reviewer #2 (Remarks to the Author):

In this study, the authors made 114 phosphatase mutants in *Cryptococcus neoformans* and analyzed their various in vitro phenotypes. They also assessed their virulence level in an insect and a mouse model. I applaud this group for their continuous effort making large strain collections (after the transcription factor and kinase deletion mutants) and make them publicly assessable. These strains and the identified phenotypes associated with each strain will be a valuable resource to the research community.

Response: We thank the reviewer for appreciating the quality of our work.

I have only a few comments/suggestions:

1. One thing that will significantly increase the value of the current work is to incorporate the current phosphatase data with previous data on TFs and kinases in the same phosphatase database. As the authors stated, the goal of this work was to find out the interplay of the phosphatase network with the TF and kinase networks. For example, if you search a certain phosphatase, you should be able to get links to kinases or TFs that have some connections to this phosphatase based on correlated phenotypes or expression profiles.

Response: We agree with this comment. Although the published TF and kinase databases are used widely and we expect that the new phosphatase database will be used widely, correlations between TFs, kinases, and phosphatases have not yet been demonstrated experimentally. We attempted to make co-phenotypic clusters of TF, kinase, and phosphatase mutants, but found that even well-established correlations between known signalling components, such as Hog1 and Ptp2 and Cna1 and Crz1, were not evident in the co-clustering analysis. There could be several explanations for this. Mutation of a phosphatase gene, which functions as a negative feedback regulator of a kinase-dependent pathway, may not lead to a phenotype opposite to the kinase mutant phenotype. For example, mutations in *PTP2*, which encodes a major negative feedback regulator of Hog1 MAPK, results in phenotypes similar to *hog1* mutant phenotypes. In addition, TF mutants generally have milder phenotypes than their upstream kinase and phosphatase mutants. For example, mutations in

CRZ1, which encodes a downstream TF activated by Cna1 phosphatase, result in much milder phenotypes than mutations in *cna1*. Thus, simple co-phenotypic clustering of TFs, kinases, and phosphatases may lead to misinterpretations of the correlations between the signalling components. Functional and mechanistic relationships between signal components should be further investigated by RNA-seq, ChIP-seq, phosphoproteomics, and protein-protein interaction assays. However, we launched a website named “*Cryptococcus neoformans* Phenome Gateway Database” (<http://www.cryptococcus.org/>) in which every TF, kinase and phosphatase gene studied is linked to the most widely used fungal genome database “FungiDB” (<https://fungidb.org/fungidb/>) to maximize connectivity between research data. This database will be updated periodically with new data and the connection to FungiDB will be maintained. This information was added to the revised manuscript as follows:

Lines 125-129 (in the Results section): We also developed the *Cryptococcus neoformans* Phenome Gateway Database (<http://www.cryptococcus.org/>) in which every TFs, kinases, and phosphatases studied^{4,5} are linked to the most widely used fungal genome database “FungiDB” (<https://fungidb.org/fungidb/>) to maximize connectivity between research data.

2. The *in vivo* expression of these genes in different organs was analyzed at DPI 3, 7, 14 and 21. However, in the inhalation infection model, there is no dissemination of *cryptococcus* from lungs to other organs by day 3. Often no dissemination could be detected even at day 7. Please explain this in detail in manuscript.

Response: As the reviewer mentioned, cryptococcal CFUs are barely detected in the brain and other organs, except the lungs, during early infection in the intranasal inhalation infection model, partly due to the detection limit of the conventional fungal burden assay. In our study, we used the NanoString nCounter platform, which can detect a single *C. neoformans* gene transcript without amplification. Thus, detection of a phosphatase transcript in the brain and/or organs other than the lungs during early infection may indicate that a small number of cryptococcal cells were released from the lungs and immune cells, disseminated to the bloodstream, and reached the brain and other organs. Based on *in vivo* real-time visualization of *C. neoformans* dissemination from the blood to the brain [Shi et al. J Clin Invest. 2010;1683-1693], intravenously injected *Cryptococcus* cells can reach the brain within a few minutes. Therefore, once a small number of *Cryptococcus* cells are disseminated from the lungs to the bloodstream at 3–7 DPI, they may reach the brain and other organs, and can be detected by NanoString analysis. As suggested by the reviewer, we revised the manuscript to describe this possibility in the Results section as follows:

Lines 157-164: Notably, we detected increased *in vivo* expression of numerous phosphatase genes in the brain, kidney, and spleen during early infection (3–7 dpi; Fig. 2). Normally, cryptococcal CFUs are barely detected in the brain and other organs, except the lungs, during early infection in the intranasal inhalation infection model, partly due to the detection limit of the conventional fungal burden assay. However, we used the NanoString nCounter platform, which can detect a single gene transcript without amplification^{99,100}. Therefore, once a small number of *C. neoformans* cells are hematogenously disseminated from the lungs to other organs during early infection, phosphatase transcripts can be detected 3–7 dpi.

3. In the *in vitro* BBB crossing assays, CFUs were counted from the bottom well after 24 hours of incubation. *Cryptococcus* replication in the bottom well will complicate the measure of fungal cells transmigrated from the top well. Similar issues exist for adhesion assays.

Response: We found that *C. neoformans* does not actively grow at 37°C in the tissue culture medium that we used for the 24 h *in vitro* blood-brain barrier (BBB) crossing and adhesion assays (please refer to the data shown below), probably due to the low glucose concentration (0.1% glucose in the tissue culture medium vs. 2% glucose in YPD). Therefore, it is unlikely that *Cryptococcus* replication in the bottom well complicated our data. Furthermore, we did not include any phosphatase mutants with growth defects at 37°C in these assays, and we determined the transmigration and adhesion levels of all mutants relative to WT levels.

Figure (for review purpose): The growth rate of *C. neoformans* in YPD medium (yeast extract-peptone-dextrose; nutrient-rich medium) and tissue culture medium (TCM) (EGM-2, Lonza Group, Basel, Switzerland) used for BBB crossing and adhesion assays

To clarify this point, we added the following sentences in the Results section.

Lines 343-346: We found that *C. neoformans* does not actively grow at 37°C in the tissue culture medium that we used for the 24 h *in vitro* BBB crossing and adhesion assays (data not shown), probably due to the low glucose concentration (0.1% glucose). Therefore, it is unlikely that *Cryptococcus* replication in the bottom well complicated our data.

Reviewer #3 (Remarks to the Author):

This manuscript reports the molecular characterization of the whole set of phosphatases in the human fungal pathogen *Cryptococcus neoformans*. The authors provided a very deep analysis of the phosphatase null mutants that surely will benefit the large community working with *C. neoformans*. Most of their findings are novel and open several new questions and research opportunities for studying signal transduction in human pathogenic fungi. That is an amazing effort to have a comprehensive understanding about the phosphatase function in this organism. The authors also try to integrate previous information (provided by their own work) about the collaboration among phosphatases, kinases, and transcription factors. This manuscript deserves publication in *Nature Communications*. I would like to provide few suggestions that are necessarily mandatory to allow the manuscript acceptance.

1) The authors failed to obtain any viable transformants for some of the phosphatases (or obtained only potential aneuploid mutants possessing both wild-type and mutant alleles), suggesting these genes may be essential. Is not possible to construct conditional mutants at least for some of these gene candidates aiming to validate their hypothesis about gene essentiality?

Response: Thank you for this constructive comment. In fact, my lab is currently validating the essentiality of putative essential TFs, kinases, and phosphatases for which we could not generate deletion mutants in this and previous studies [Jung et al. 2015 *Nat Commun*;6757; Lee et al 2016 *Nat Commun*;12766]. However, this requires a series of sophisticated analyses, including construction of conditional mutants using the copper-regulated *CTR4* promoter and sporulation analysis of heterozygous mutants in diploid *C. neoformans* backgrounds. Even if the growth of a conditional mutant of a putative essential gene was retarded under the repressed condition, it does not validate its essentiality because it could be growth-required rather than essential. For genes that are verified by conditional gene expression and sporulation analysis of heterozygous mutants, we are constructing histone 3 (H3) promoter-based overexpression strains to characterize their functions. Although reporting the essentiality and related functions of the putative phosphatase genes will be extremely valuable and informative, as the reviewer suggested, we respectively submit that these studies are beyond the scope of the current study and may be presented in another manuscript.

2) there was some discussion about the influence of calcineurin in the regulation of HAD1. I noticed that there were no experiments in Figure 3 with calcineurin inhibitors like cyclosporine or FK506. Also there was no experiment with rapamycin. These drugs are important to implicate phosphatases in the function of calcineurin or TOR.

Response: Phosphoproteomic analysis of WT *C. neoformans* and the *cna1* deletion mutant after treatment with FK506 demonstrated that Had1 is a calcineurin target [Park et al. *PLoS Pathog*.

2016;e1005873], and the pathogenicity-related functions of Had1 have been published by Jung et al [Jung et al. G3 (Bethesda). 2018; 643-652]. Based on these data, we described Had1 as a downstream target gene of the calcineurin pathway. For clarification, we revised the Discussion as follows:

Lines 461-468: The pathobiological functions of Ymr1, Cdc1, Had1, and Pph3 remain unclear because mutants of these genes do not demonstrate clear phenotypic traits related to *C. neoformans* pathogenicity. However, the *ymr1*Δ, *cdc1*Δ, and *had1*Δ mutants showed increased susceptibility to cell membrane disruption, and this instability may have contributed to reduced infectivity or virulence relative to WT. The *had1*Δ mutant also exhibited increased susceptibility to cell wall-destabilizing agents, which is in good agreement with recent reports showing that Had1 is a potential target of calcineurin, which plays a role in regulating cell wall integrity⁷³. Had1 was also shown to be required for virulence in a murine model of systemic cryptococcosis¹⁶.

As the reviewer suggested, data regarding the susceptibility/resistance of *C. neoformans* phosphatase mutants to FK506 and rapamycin are needed to understand the comprehensive signalling networks governed by the calcineurin and TOR pathways, which play critical roles in growth, development, the stress response, and virulence of *C. neoformans*. Our lab has been working on these pathways along with our collaborators for a long time, and we are currently investigating the susceptibility/resistance of all our TF, kinase, and phosphatase mutants to FK506 and rapamycin. Although we identified a number of mutants exhibiting altered susceptibility to FK506, rapamycin, or both, we still need more experimental data to validate the mechanistic relationships of these genes with the calcineurin and/or TOR pathways. Therefore, we respectfully submit that these studies are beyond the scope of the current study.

3) It could be interesting some speculation about the number of phosphatase regulatory subunits present in *C. neoformans* genome.

Response: We agree and thank the reviewer for this constructive comment. Characterization of phosphatase regulatory subunits is critical for development of novel drugs that target phosphatases, such as FK506, which targets the catalytic and regulatory subunits of the calcineurin complex via FKBP12. Because only a few studies on fungal phosphatase regulatory subunits have been published, we searched for putative phosphatase regulatory subunits by BLASTp analysis using annotated information from the *Saccharomyces* Genome Database (SGD, <http://yeastgenome.org>). Using forward and reverse BLASTp analyses, we found 26 putative phosphatase regulatory subunits in *C. neoformans*. However, it remains unclear whether these are authentic phosphatase regulatory subunits. Further research is needed to unravel the mechanistic and functional relationships between phosphatases and their putative regulatory subunits. We merged this additional information with the section on the development of antifungal drugs that target phosphatases and/or phosphatase regulatory subunits (as suggested by reviewer 4) in the Discussion section as follows:

Lines 525-534: To increase the specificity of a drug that targets a phosphatase, both the catalytic and regulatory subunits of the phosphatase complex should be considered, such as was considered with the drug FK506, which targets both the catalytic and regulatory subunits of the calcineurin complex via FKBP12⁹⁴. Based on BLASTp analysis using annotated information from the *Saccharomyces* Genome Database (SGD, <http://yeastgenome.org>), we found 26 putative phosphatase regulatory subunit orthologues in *C. neoformans* (Supplementary Data 10). However, further research is needed to characterise the functional and mechanistic relationships between the catalytic and regulatory subunits of phosphatases; this research could lead to the development of more antifungal drugs that target phosphatases.

4) An annoying detail: Mre11 is the name for a nuclease subunit of the MRX complex with Rad50p and Xrs2p. This complex functions in repair of DNA double-strand breaks and in telomere stability. Is it not possible to rename the phosphatase Mre11 ?

Response: We named all of the phosphatases according to the published nomenclature guideline [Inglis et al Eukaryot Cell. 2014;878-83]. Mre11 was retrieved and named as a result of BLASTp analysis with ScMre11 (Score 328, e-value 5e⁻¹⁰³). Reverse BLASTp from cryptococcal Mre11 to

ScMre11 was confirmed (Score 227.2; e-value $1.8e^{-63}$). We also found that *mre11Δ* mutants were more susceptible to hydroxyurea and methyl methanesulfonate than WT, suggesting that Mre11 may be involved in DNA damage repair. We included Mre11 as a putative phosphatase because it has a metallo-dependent phosphatase domain. The function of the putative phosphatase domain in Mre11 should be studied in the future. Similarly, Cac1, a well-known adenylyl cyclase that produces cAMP, was included as a putative phosphatase because it contains a phosphatase domain.

Reviewer #4 (Remarks to the Author):

This manuscript by Jin et al performed systematic analyses on the phosphatase mutants in the human fungal pathogen *Cryptococcus neoformans*. This is a very comprehensive study with solid data.

Response: We thank the reviewer for appreciating the quality of our work.

However, given this is the 4th -omic study from this group, the authors should have better dataset summarization and organization rather than simply listing each phenotypic score or each growth curve. Also, if the authors wanted to emphasize the potential drug target value of certain phosphatases, it is better to include the discussion of feasibility of drug targeting, for example, to compare the similarity between fungal phosphatase and human's. There are a lot of conserved essential genes across fungal species, but not suitable as drug target at all.

Response: We thank the reviewer for this critical comment. As suggested by the reviewer, we replaced the original Figure 3c (all these data can be seen in the new Figure 2) with BLAST matrix data for 31 pathogenicity-related phosphatases to highlight the evolutionary relationships between the orthologous phosphatases. We found that five of the phosphatases do not have evident orthologues in humans (Hs in the new Figure 3c): Tps2, Siw14, Had1, Oca101, and Oca1. Thus, these five pathogenicity-related phosphatases could be excellent anticryptococcal targets. In particular, Tps2, Had1, and Oca1 are required for *C. albicans* virulence [Zaragoza et al. Microbiology 2002;1281-90; Desai et al. Pathogens 2015;573-89; Hanaoka et al. Eukaryot Cell 2008;1640-8], thus drugs that target these phosphatases could have broad antifungal activity. The trehalose pathway consisting of Tps1 (trehalose-6-phosphate synthase) and Tps2 (trehalose-6-phosphate phosphatase) has been highlighted as a promising antifungal drug target because mammals do not have an equivalent pathway [Perfect et al. Virulence 2017;143-149]. Recently, the structures of the N-terminal domain of *C. albicans* Tps2 and *S. cerevisiae* inositol phosphatase Siw14 have been resolved [Miao et al. PNAS USA 2016;7148-53; Wang et al. J Biol Chem 2018;6905-6914], but there is no structural information on Had1 and Oca1/Oca101 yet. Current and future structural studies will provide a structural basis for the design of novel antifungal drugs that target phosphatases. In addition, an evolutionarily conserved pathogenicity-related phosphatase could be used as an antifungal drug target if the drug is developed with increased specificity to fungal orthologs than to human orthologs. This has been done with Cna1 in the calcineurin pathway. Cna1 can be inhibited by FK506, a well-known immunosuppressive drug, through FK506-binding proteins. Recently, several research groups (including us) have generated FK506 analogues with greatly reduced immunosuppressive activity but significant antifungal activity [Lee et al. Antimicrob Agents Chemother. 2018;e01627-18; Nambu et al. Bioorg Med Chem Lett. 2017;2465-2471; Odom et al. Antimicrob Agents Chemother. 1997; 156-61]. This information has been added to the Results and Discussion sections as follows:

Lines 204-208: Of the 31 pathogenicity-related phosphatases identified, five do not have evident orthologues in humans (Hs in Fig 3c): Tps2, Siw14, Had1, Oca101, and Oca1. Therefore, these five pathogenicity-related phosphatases could be excellent anticryptococcal targets. Tps2, Had1, and Oca1 are also required for the virulence of *C. albicans*²⁵⁻²⁷, thus drugs that target these phosphatases could have broad antifungal activity.

Lines 507-525: The 31 pathogenicity-related phosphatases that were identified here and in previous studies could be potential antifungal drug targets, especially Sit4, Cna1, and Tps2, which are required for virulence of the major human fungal pathogens *C. neoformans*, *C. albicans*, and *A. fumigatus*. Sit4, Yvh1, Sdp101, Ptp2, Cac1, Tps2, Inp5201, and Ppg1 are required for virulence of both human and plant fungal pathogens, thus could be targets for the development of broad-spectrum antifungal drugs. Tps2 is a particularly promising antifungal drug target because the trehalose pathway is missing in

humans⁸⁸. Recently, the structure of the N-terminal domain of *C. albicans* Tps2 has been resolved⁸⁹ and provides a structural basis for the design of Tps2-specific antifungal agents. Siw14, Had1, and Oca1/101 could also be good targets for cryptococcal treatment because there are no evident orthologues of these phosphatases in humans. Recently, the structure of the *S. cerevisiae* inositol phosphatase Siw14 has also been resolved⁹⁰, whereas structural information for Had1 and Oca1/Oca101 is not yet available.

Several clinical strategies have targeted phosphatases directly by targeting the catalytic subunit of phosphatase complexes or indirectly by targeting the regulatory or scaffolding subunits of phosphatase complexes. Examples of the former strategy include LB-100 inhibition of the PP2A-C subunit⁹¹, FK506 inhibition of the calcineurin complex⁹²⁻⁹⁴, and inhibition of dual-specificity phosphatases by several small molecules⁹⁵. An example of the latter strategy includes ceramide and its derivatives, which are bioactive sphingolipid molecules that activate PP2A by preventing SET inhibitor binding to PP2A^{96,97}.

Figure 3 (c) A BLAST matrix comparative search for pathogenicity-related phosphatases was performed using the Comparative Fungal Genomics Platform (<http://cfgp.riceblast.snu.ac.kr>). Abbreviations: Pi, *Phytophthora infestans*; Af, *Aspergillus fumigatus*; An, *Aspergillus nidulans*; Bg, *Blumeria graminis*; Bc, *Botrytis cinerea*; Ci, *Coccidioides immitis*; Cg, *Colletotrichum graminicola*; Fg, *Fusarium graminearum*; Fo, *Fusarium oxysporum*; Hc, *Histoplasma capsulatum*; Mo, *Magnaporthe oryzae*; Mg, *Mycosphaerella graminicola*; Nc, *Neurospora crassa*; Pa, *Podospora anserine*; Ca, *Candida albicans*; Sc, *Saccharomyces cerevisiae*; Sp, *Schizosaccharomyces pombe*; Cn, *Cryptococcus neoformans*; Hi, *Heterobasidion irregular*; Lb, *Laccaria bicolor*; Pc, *Phanerochaete chrysosporium*; Sl, *Serpula lacrymans*; Ml, *Melampsora laricis-populina*; Pg, *Puccinia graminis*; Um, *Ustilago maydis*; Am, *Allomyces macrogynus*; Bd, *Batrachochytrium dendrobatidis*; Ec, *Encephalitozoon cuniculi*; Pb, *Phycomyces blakesleeanus*; Ro, *Rhizopus oryzae*; Dm, *Dorosophila melanogaster*; Hs, *Homo sapiens*; Ce, *Caenorhabditis elegans*; At, *Arabidopsis thaliana*; Os, *Oryza sativa*.

Specific points:

Figure 1C: Please make sure the pie charts of each species have the same size and format.

Response: We adjusted the sizes and formats of the pie charts as suggested.

Figure 2: It's fine to display the homology level of each phosphatase, but not very informative for the central question of the study. The authors clustered the kinase mutants according to the phenotypic assays in their previous work, that's a very nice functional analysis. Can such kind of analysis be performed here?

Response: We thank the reviewer for this comment. As suggested, we moved the original Figure 2 to Supplemental Figure 2 and made a new Figure 2 showing phenotypic clustering of 60 phosphatase mutants with their corresponding *in vivo* gene expression levels measured by NanoString analysis. We also added clustering of NanoString data in Supplementary Fig. 6. As the reviewer mentioned, in our previous report [Lee et al. Nat Commun. 2016;12766], phenotypic clustering of kinase mutants allowed for identification of groups of kinases belonging to the same signalling pathway because kinases in a group are serially activated in a given signalling pathway. For example, a MAPK pathway is activated by a MAPK kinase (MAPKK), which is activated by a MAPKK kinase (MAPKKK). In contrast, it is rare for a series of phosphatases to be activated as a cascade in the same signalling pathway. However, there could be functional correlations among a group of phenotypically clustered phosphatases. Based on this new clustering data, we revised the text as follows:

Lines 144-146: Phenotypic clustering of phosphatase mutants revealed groups of phosphatases that could be directly or indirectly correlated with regards to cellular function (Fig. 2).

Figure 2. Phenotypic clustering and *in vivo* gene expression profiling of phosphatases in *C. neoformans*. *In vitro* phenotypic traits were examined under 30 different growth conditions and scored on a 7-point scale (-3: strongly reduced/susceptible, -2: moderately reduced/susceptible, -1: weakly reduced/susceptible, 0: WT-like, +1: weakly enhanced/tolerant, +2: moderately enhanced/tolerant, +3: strongly enhanced/tolerant). All phenotypic data are available in the *Cryptococcus neoformans* Phosphatase Phenome Database (<http://phosphatase.cryptococcus.org>).

More than three biologically independent experiments were performed for each phenotypic trait. Hierarchical phenotypic clustering of 60 phosphatases showing at least one phenotypic trait was performed with one minus Pearson correlation in Morpheus (<https://software.broadinstitute.org/morpheus>). The right panel shows the corresponding *in vivo* gene expression profiles for each phosphatase gene determined by NanoString nCounter platform analysis during intranasal murine infection with *C. neoformans*. Red letters represent pathogenicity-related phosphatases. The $\Delta sdp101$ mutant, which did not show any *in vitro* phenotypes but exhibited reduced infectivity, was also included. Abbreviations: 25, 25°C; 30, 30°C; 37, 37°C; 39, 39°C; CAP, capsule production; MEL, melanin production; URE, urease production; MAT, mating; HPX, hydrogen peroxide; TBH, tert-butyl hydroperoxide; MD, menadione; DIA, diamide; MMS, methyl methanesulphonate; HU, hydroxyurea; 5FC, 5-flucytosine; AMB, amphotericin B; FCZ, fluconazole; FDX, fludioxonil; TM, tunicamycin; DTT, dithiothreitol; CDS, cadmium sulphate; SDS, sodium dodecyl sulphate; CR, Congo red; CFW, calcofluor white; KCR, YPD+1.5 M KCl; NCR, YPD+1.5 M NaCl; SBR, YPD+2 M sorbitol; KCS, YP+1 M KCl; NCS, YP+1 M NaCl; SBS, YP+2 M sorbitol.

Supplementary Figure 6. Clustering based on *in vivo* gene expression of *C. neoformans* phosphatases. *In vivo* gene expression profiles of *C. neoformans* phosphatases were hierarchically clustered using one minus Pearson correlation in Morpheus (<https://software.broadinstitute.org/morpheus>). The original *in vivo* gene expression data obtained from the NanoString nCounter analysis are available in Supplementary Data 8. Of the 139 phosphatases, the pathogenicity-related phosphatases are given in the left panel. The groups highlighted in the right panel include the phosphatase genes that exhibited high *in vivo* expression in the lungs, brain, and/or kidney. Red letters indicate the pathogenicity-related phosphatases and green letters indicate putative essential phosphatases in *C. neoformans*.

Figure 3A: please label number of larvae each test group used. And please double check the p value of each test, for example, in the panel of GDA1, the orange curve is closer to the black curve (WT control), but the p value of orange curve is smaller than the pink one.

Response: We agree and we labelled the number of larvae in each test group as suggested. We apologize that the *P* values for the two independent *gda1* mutants were accidentally switched, and we corrected this mistake.

Figure 3. Pathogenicity-related phosphatases and their phenotypic traits in *C. neoformans*. (a) Virulence-regulating phosphatases were identified by a *Galleria mellonella* insect killing assay ($n \geq 15$). *P* values were calculated using the log-rank test to measure statistical differences between the WT strain (H99S) and phosphatase mutants.

Figure 3C: For the phenotypic traits evaluation, are these the sub-dataset from Figure 2? It is redundant data presentation if no new information provided. Also, is there any particular reason using different size of dot other than the color gradient?

Response: As previously suggested by the reviewer, we replaced the original Figure 2 (moved to Supplemental Figure 3) with phenotypic clustering of 60 phosphatase mutants, which showed at least one phenotypic trait, and their corresponding *in vivo* gene expression levels determined by NanoString analysis (please refer to our response above). The $sdp101\Delta$ mutant, which did not display an *in vitro* phenotype but exhibited reduced infectivity, was also included. To eliminate redundancy, we replaced the original Figure 3c (all these data can be seen in the new Figure 2) with BLAST matrix data for 31 pathogenicity-related phosphatases to highlight the evolutionary relationships between the orthologous proteins. This new Fig. 3c also addresses the reviewer's major comment regarding fungal and human phosphatase comparisons for the development of novel antifungal drugs that target phosphatases.

Figure 4-6: It is better to group the virulence traits assays in one figure, with representative mutants.

Response: We agree with this comment. We combined the core data of three figures (growth at high temperature, capsule production, and melanin synthesis) into Fig. 4 and moved the remaining data into Supplementary Figures 8-10.

Lines 210-227: We next focused on the pathobiological functions of the 31 pathogenicity-related phosphatases in *C. neoformans*. First, because thermotolerance for mammalian body temperatures is a critical virulence factor for most human fungal pathogens, we quantitatively measured growth of each mutant at 30°C and 37°C. The *gua1Δ*, *yvh1Δ*, *fbp26Δ*, *siw14Δ*, *dbr1Δ*, *ccr4Δ*, *ppg1Δ*, *nem1Δ*, and *inp5201Δ* mutants showed impaired growth at both 30°C and 37°C (Fig. 4a and Supplementary Fig. 8). Of these, the *ccr4Δ*, *ppg1Δ*, *nem1Δ*, *dbr1Δ*, and *inp5201Δ* mutants exhibited more growth defects at 37°C than 30°C. The *ssu72Δ*, *phs1Δ*, *mre11Δ*, *tps2Δ*, and *cna1Δ* mutants showed impaired growth at 37°C but not at 30°C. A total of 14 phosphatase mutants showed impaired growth at 37°C relative to WT (Fig. 4a and Supplementary Fig. 8) and showed reduced murine infectivity or insect virulence relative to WT (Fig. 3). The *ppg1Δ*, *cna1Δ*, and *tps2Δ* mutants exhibited the most significant growth defects at 37°C and did not grow to the level of the WT even after an extended incubation period (Fig. 4a). Concordantly, *ppg1Δ*, *cna1Δ*, and *tps2Δ* mutants exhibited highly reduced lung and brain signature-tagged mutagenesis (STM) values (< -5 ; Fig. 3b). The *oca101Δ* mutant showed impaired growth relative to WT at 30°C but not at 37°C (Supplementary Fig. 8) suggesting that the role of Oca101 in *C. neoformans* pathogenicity is not related to temperature. Collectively, these data suggest that growth at 37°C is a critical for virulence of *C. neoformans*.

Figure 4. Phosphatases involved in *C. neoformans* virulence. (a) Growth curves of WT and phosphatase mutants were generated at 30°C (control) and 37°C (mammalian body temperature). Fifteen phosphatase mutants had growth defects at 30°C and 37°C. Nine of these phosphatase mutants had more substantial growth defects at 37°C than 30°C (additional data in Supplementary Fig. 8). Each curve represents data from two independent experiments (see Supplementary Fig. 8 for data from the individual experiments). Optical density at 600 nm (OD_{600nm}) was measured with a multi-channel bioreactor (Biosan Laboratories, Inc., Warren, MI, USA) for 40–90 h based on growth rate. (b) Melanin production was measured using three different melanin-inducing media (Niger seed, dopamine, and epinephrine medium). Representative images from three independent experiments are shown here. Each strain was spotted on medium containing 0.1% glucose, incubated at 30°C, and photographed after 1–3 days. (c-d) Gene expression of the melanin-regulating genes *LAC1*, *BZP4*, and *HOB1* were determined by qRT-PCR in both nutrient-rich (R) and nutrient-starvation (S) conditions. RNA was extracted from three biological replicates with three technical replicates of WT and melanin-regulating phosphatase mutants. Expression was normalised to *ACT1*, and statistical significance was calculated by one-way ANOVA with the Bonferroni multiple comparisons test ($*P < 0.05$, $**P < 0.001$, $***P < 0.0001$). Error bars represent standard error of the mean. (e) The capsule production assay was performed using capsule-inducing media (FBS agar medium). Capsule thickness (total diameter – cell body diameter) was measured for WT cells ($n = 50$) and for each phosphatase mutant ($n = 50$). Statistical significance was calculated by one-way ANOVA with the Bonferroni multiple comparisons test ($*P < 0.05$, $**P < 0.001$, $***P < 0.0001$). Error bars indicate standard deviation. The graph is representative of more than three independent experiments. The images are representative DIC images of WT and phosphatase mutants incubated on FBS agar medium and stained with India ink. Scale bars, 10 μ m.

Figure 8: The authors showed the abnormal mannosylation of GDA1 and YND1 mutation, however, they did not really establish the relationship between mannosylation and virulence in these two mutants. With current data, it is not appropriate to state “xxx modulate virulence of *C. neoformans* by controlling O-mannosylation”.

Response: We agree with this comment. We revised the sentence as follows:

Line 294-295: Two virulence-related apyrases, Gda1 and Ynd1, modulate O-mannosylation in *C. neoformans*.

Figure 9: I wonder if the in vitro BBB traversing data could correlate with in vivo results, by intravenously infect mice with the mutants and check the brain level. In vitro assay results need to be confirmed by in vivo data

Response: Thank you for this constructive comment. We employed the suggested strategy to identify TFs and kinases that govern brain infection by *C. neoformans* in a recently published study [Lee et al. Nat Commun. 2020;1521]. In this study, we compared lung STM scores of mice intranasally infected with *C. neoformans* TF and kinase mutants with brain STM scores of mice infected with the same *C.*

neoformans TF and kinase mutants by intravenous injection, which bypasses the lungs, or intracerebroventricular injection, which bypasses the lungs and the BBB. We identified TFs and kinases involved in crossing the BBB and/or survival in brain parenchyma. Notably, we found that a number of TFs and kinases are specifically involved in *C. neoformans* brain infection, but not lung infection. In future studies, we plan to employ the same strategy to systematically identify phosphatases and their regulatory subunits that govern brain infection by *C. neoformans* and to correlate the phosphatase data with the brain infection-related kinase and TF data. We respectively submit that these studies are beyond the scope of the current study.

Figure 10: The idea of “core pathogenicity-related phosphatases” is not very clear. What specific pathways or functions related to these phosphatases make them broadly related to virulence (or actually in vivo fitness)? The authors should at least have more discussion on this instead of purely listing the overlapped or not overlapped genes.

Response: We agree with this comment. We listed Sit4, Cna1/Cmp1/CnaA, Tps2, Yvh1, Nem1, Cac1/Cyr1/Mac1, Sdp101/Pmp1/Spd2/Msg5, Vps29, and Ppg1 as common pathogenicity-related phosphatases in the major human fungal pathogens *C. neoformans*, *C. albicans*, and *A. fumigatus*, and in the plant fungal pathogens *M. oryzae* and *F. graminearum*, and we refer to these as core pathogenicity-related phosphatases. We further discussed the functions and signalling pathways of these core pathogenicity-related phosphatases as follows:

Line 371-393: In the human yeast pathogens *C. neoformans* and *C. albicans*, the following 13 phosphatases are considered core pathogenicity-related phosphatases: Cna1/Cmp1, Sit4, Oca1, Yvh1, Sdp101/Cpp1, Ptp2/Ptp3, Cac1/Cyr1, Ccr4, Had1/Rhr2, Tps2, Inp5201/Inp51, Ppg1, and Gua1. Of the 13 phosphatases, CnaA, SitA, and OrlA (a Tps2 ortholog) have been shown to be required for *A. fumigatus* virulence⁵³⁻⁵⁵. Upon comparison with pathogenicity-related phosphatases in *F. graminearum*, eight phosphatases have been shown to be required for the virulence of both animal and plant fungal pathogens: Sit4, Yvh1, Sdp2/Msg5 (an Sdp101 orthologue), Ptp2, Ac1 (a Cac1 orthologue), Tps2, Inp53 (an Inp5201 orthologue), and Ppg1. Sit4 is involved in the TOR pathway, Cac1 is involved in the cAMP pathway, and Ppg1 and Yvh1^{51,66-68} are involved in cell growth, nutrient sensing, and the stress response in fungal pathogens^{2,69}. The Cna1 and Had1-mediated calcineurin pathway, the Tps2-mediated trehalose pathway, and the Msg5-mediated Mpk1/Sit2 MAPK pathway are all required for maintaining cell wall integrity. Ptp2, which is a major negative feedback regulator of the HOG pathway, is involved in adaptation and the stress response in fungal pathogens. Inp51, Inp52, and Inp53 are involved in phosphoinositide signalling, which controls vesicle trafficking, the actin cytoskeleton, and cell wall integrity^{70,71}. Based on these data, phosphatases and signalling pathways involved in cell growth, nutrient sensing, cell wall integrity, the stress response, and phosphoinositide signalling appear to play pivotal roles in general fungal pathogenicity. Notably, however, deletion of *PPH3* reduces *C. neoformans* and *F. graminearum* virulence but enhances *C. albicans* virulence^{8,42}. In contrast, deletion of *PTC2* and *PTC3* reduces *C. albicans* and *F. graminearum* virulence, respectively, but does not reduce *C. neoformans* virulence^{27,72}. Thus, some phosphatases may play differential roles in controlling the virulence of various fungal pathogens.

REVIEWERS' COMMENTS:

Reviewer #2 (Remarks to the Author):

It would be important to point out in the main text that there is no correlations between TFs, kinases, and phosphatases based on the experimental data present in this and previous studies. The authors provided such information in the response letter, and should include this in the main text.

Reviewer #4 (Remarks to the Author):

The authors have addressed all my comments and revised the manuscript and figures properly.

Response letter to referees

REVIEWERS' COMMENTS:

Reviewer #2 (Remarks to the Author):

It would be important to point out in the main text that there is no correlations between TFs, kinases, and phosphatases based on the experimental data present in this and previous studies. The authors provided such information in the response letter, and should include this in the main text.

Response: We agree with this concern. So, we added these comments in the Discussion section.

Line 474-Line 488: To identify functional correlations between TFs, kinases, and phosphatases, we attempted to make co-phenotypic clusters of TF, kinase, and phosphatase mutants constructed by this and previous studies^{4,5}, but found that even well-established correlations between known signalling components, such as Hog1 and Ptp2 and Cna1 and Crz1, were not evident in the co-clustering analysis. There could be several explanations for this. Mutation of a phosphatase gene, which functions as a negative feedback regulator of a kinase-dependent pathway, may not lead to a phenotype opposite to the kinase mutant phenotype. For example, mutations in *PTP2*, which encodes a major negative feedback regulator of Hog1 MAPK, results in phenotypes similar to *hog1Δ* mutant phenotypes^{10,73}. In addition, TF mutants generally have milder phenotypes than their upstream kinase and phosphatase mutants. For example, mutations in *CRZ1*, which encodes a downstream TF activated by Cna1 phosphatase, result in much milder phenotypes than mutations in *CNA1*⁷⁴. Thus, simple co-phenotypic clustering of TFs, kinases, and phosphatases may lead to misinterpretations of the correlations between the signalling components. Functional and mechanistic relationships between signal components should be further investigated by RNA-seq, ChIP-seq, phosphoproteomics, and protein-protein interaction assays in future studies.

Reviewer #4 (Remarks to the Author):

The authors have addressed all my comments and revised the manuscript and figures properly.

Response: We appreciate this.